# T-Cell Receptor Sequences Identify Combined Coxsackievirus–*Streptococci* Infections as Triggers for Autoimmune Myocarditis and Coxsackievirus–*Clostridia* Infections for Type 1 Diabetes

**DOI:** 10.3390/ijms25031797

**Published:** 2024-02-01

**Authors:** Robert Root-Bernstein

**Affiliations:** Department of Physiology, Michigan State University, East Lansing, MI 48824, USA; rootbern@msu.edu

**Keywords:** T-cell receptors, autoimmunity, mimicry, anti-idiotype, antigen complementarity, synergism, autoimmune myocarditis, diabetes, coxsackievirus, *Streptococci*, *Clostridia*, insulin, myosin, laminin

## Abstract

Recent research suggests that T-cell receptor (TCR) sequences expanded during human immunodeficiency virus and SARS-CoV-2 infections unexpectedly mimic these viruses. The hypothesis tested here is that TCR sequences expanded in patients with type 1 diabetes mellitus (T1DM) and autoimmune myocarditis (AM) mimic the infectious triggers of these diseases. Indeed, TCR sequences mimicking coxsackieviruses, which are implicated as triggers of both diseases, are statistically significantly increased in both T1DM and AM patients. However, TCRs mimicking *Clostridia* antigens are significantly expanded in T1DM, whereas TCRs mimicking *Streptococcal* antigens are expanded in AM. Notably, *Clostridia* antigens mimic T1DM autoantigens, such as insulin and glutamic acid decarboxylase, whereas *Streptococcal* antigens mimic cardiac autoantigens, such as myosin and laminins. Thus, T1DM may be triggered by combined infections of coxsackieviruses with *Clostridia* bacteria, while AM may be triggered by coxsackieviruses with *Streptococci.* These TCR results are consistent with both epidemiological and clinical data and recent experimental studies of cross-reactivities of coxsackievirus, *Clostridial*, and *Streptococcal* antibodies with T1DM and AM antigens. These data provide the basis for developing novel animal models of AM and T1DM and may provide a generalizable method for revealing the etiologies of other autoimmune diseases. Theories to explain these results are explored.

## 1. Introduction

### Problems and Hypotheses

The natural causes of human autoimmune diseases have yet to be discovered despite decades of epidemiological and experimental studies. While genetic predispositions are associated with susceptibility to most autoimmune diseases, infectious triggers are also thought to be necessary but have thus far resisted unambiguous elucidation. Recent research has unexpectedly found that T-cell receptor (TCR) sequences expanded during various infections, such as those caused by human immunodeficiency virus (HIV) and SARS-CoV-2, mimic these viral antigens rather than being complementary to them [1,2,3,4,5,6,7,8]. This mimicry is evident in three types of observations: (1) similarity searches reveal at least 60% identities of the amino acids in sequences of ten or more amino acids; (2) these similarities occur at statistically significantly increased rates compared to randomly chosen TCR sequences; and (3) antibodies against protein sequences mimicked by TCRs also bind to the TCRs, demonstrating that the similarities are antigenically significant. These observations suggest that it may be possible to use TCR sequences derived from expanded T cells to identify the triggering agents involved in diseases with unknown etiologies, such as those caused by autoimmune mechanisms. This paper tests this hypothesis with specific regard to type 1 diabetes mellitus (T1DM) and autoimmune myocarditis (AM). These two autoimmune diseases were chosen not only because of the availability of TCR sequences but also because they share a number of putative microbial triggers. As is discussed in more detail below, these possible triggers include coxsackieviruses, Epstein–Barr virus (EBV), and cytomegalovirus (CMV). Each of these viruses has also been proposed to be triggers of several other ADs, as well (see below). The association of any given microbe to multiple ADs raises the often-ignored problem of how a single microbe can cause a unique infectious disease, as well as participate in triggering several different ADs. The specific TCR sequences expanded in each AD may help to elucidate this conundrum.

An interplay of genetic and infectious factors has been implicated in both diabetes and myocarditis (e.g., [9,10,11,12,13,14]). Both viral and bacterial infections are associated with triggering T1DM, including the coxsackie viruses (CV) (both A and B strains) [15,16,17,18,19,20], other enteroviruses [21], rubella, mumps, rotaviruses, cytomegalovirus (CMV), Epstein–Barr virus (EBV), and hepatitis C virus (HCV) [11,22,23,24,25,26]. Also, Mycobacterium species and Bordetella *pertussis* [27,28], *Staphylococci* [29,30], and *Helicobacter pylori* [31] have also been reported to occur unusually frequently preceding the onset of T1DM cases, and significant differences in gut microbiota may play a critical role in the initiation or development of diabetes [32,33,34,35,36,37,38].

The diversity of possible infectious triggers unfortunately leaves many questions as to the sufficiency and necessity of any of them [23,39]. Furthermore, it has thus far proven impossible to create animal models of T1DM by using individual infectious agents from the list above. Coxsackieviruses and CMV exacerbate or accelerate pre-existing spontaneous T1DM in rodents (e.g., NOD mice) and in animals pretreated with streptozotocin [39,40] but do not trigger T1DM by themselves. Similarly, coxsackie B virus types 3 and 4 have repeatedly failed to induce diabetes in monkeys [41]. Additionally, rather than initiating disease, pertussis vaccine protects streptozotocin-treated CD-1 mice from developing diabetes [42]; immunization with *M. leprae* prevents diabetes in NOD mice [43]; and immunization with Bacillus Calmette–Guerin (BCG) NOD mice from developing diabetes [44], even improving diabetic control [45]. In short, mono-infectious approaches to modeling T1DM based on correlations between individual infections and onset of the disease have failed [46]. Thus, Filippi and von Herrath [47] proposed, “This could be explained by the fact that viral association with T1D will likely be multifactorial”.

The range of infectious triggers of autoimmune myocarditis (AM) appears to be as diverse as for T1DM. Many cases of AM are associated, like T1DM, with coxsackievirus (reviewed in [48,49]. Other microbes, including varicella zoster virus [50], cytomegalovirus [51], HCV [52], adenovirus [53], Epstein–Barr virus, parvovirus B19 [54], ECHO virus or several of these simultaneously [55,56], smallpox [57], and *Trypanosoma cruzi* [58] have all been associated with AM. If one considers rheumatic heart disease (RHD) as a form of autoimmune myocarditis, then there is also a strong association with group A and group G streptococci (GAS and GGS) (reviewed in [59,60,61]). Streptococcal vaccines are also (rarely) associated with inducing AM [62].

Many of these associations, however, have been questioned. Persistent viral infection is therefore an unlikely cause of AM [55,63]. Also confusing is the observation that the same increased serotype-specific titers of antibodies to coxsackie B3 virus were found in household members of patients, as in the AM patients themselves [64], suggesting that CVB3 may have been present and even necessary, but was not sufficient, to induce AM. Animal models would also seem to argue against unifactorial causes of AM. CVB3 only induces AM in mice when co-inoculated with cardiac myosin or with heart antigens after passaging through heart tissue [65,66,67]. CMV induces AM only when co-inoculated with salivary gland tissue [68], which is rich in actomyosin [69]. The M protein of GAS, which mimics cardiac myosin, only induces RHD when co-inoculated with a mycobacterial adjuvant [61,67,70,71]. While many investigators ignore the need for tissue or adjuvant material in describing the pathogenesis of AM/RHD, the necessity for these additional antigens is important evidence that CV infection is not sufficient to induce autoimmune disease.

As with T1DM, the multiplicity of disease agents associated with AM, as well as the need to use additional antigens in animal models, has led to speculation that AM has a multifactorial etiology, an idea first proposed in 1972 by Burch and Giles [72] (see also [73]). One likely combination is CV with GAS [73]. Epidemiological studies demonstrate that 65–95 percent of RHD and AM cases presented with concurrent GAS and CV infections, while uncomplicated CV or GAS infections were almost never associated with the development of RHD or AM [74,75,76,77,78,79,80,81]. In addition, some cases of AM are associated with multiple viral infections. Kuhl et al. [82] found that at least 25% of patients with myocarditis had evidence of more than one viral infection in biopsy specimens. In a study by Mahfoud et al. [55], 13 of 32 patients with AM confirmed by biopsy were positive for two or more different viral genomes. Andreoletti et al. [56] similarly found that 12 percent of biopsy-confirmed AM cases had viral co-infections, often involving parvovirus B19 with HHV6. Possible bacterial co-infections were not considered in these viral studies; their presence could make the etiology of AM even more complicated.

Comparing the probable infectious triggers of T1DM to AM reveals one final problem with mono-infectious theories of autoimmunity, which is that some of the infectious agents are putative triggers for both diseases. The shared microbes include coxsackie viruses, Epstein–Barr virus (EBV), and cytomegalovirus (CMV). These viruses are associated with other autoimmune diseases as well. Coxsackieviruses have been associated with the risk of amyotrophic lateral sclerosis [83] and Sjogren’s syndrome [84,85,86]. Both EBV and CMV are associated with the onset of multiple sclerosis [87] and idiopathic thrombocytopenia purpura [88,89]. EBV is additionally linked to systemic lupus erythematosus, rheumatoid arthritis, and Sjögren’s syndrome [90,91]. CMV is also correlated with the onset of systemic lupus erythematosus [92], Guillain–Barré syndrome [93], and bullous pemphigoid [94]. But how can one infectious agent trigger such very different autoimmune diseases? One possibility proffered by multifactorial theories is that different combinations of microbes induce different autoimmune responses targeted at different tissues or organs. Thus, coxsackie virus paired with, say, *Streptococcal* infection may lead to AM [73,95,96,97], whereas coxsackievirus paired with a different infectious agent may lead to diabetes [1,98]. Similarly, CMV combined with one bacterial co-infection might lead to thrombocytopenia [89], with another co-infection to MS, and with yet a third co-infection to lupus. One of the purposes of this paper is to explore this possibility with specific regard to AM/RHD and T1DM.

## 2. Results

To test the hypothesis that TCR sequences identify the infectious triggers of autoimmune diseases, two proteomic computational algorithms were employed: BLAST and LALIGN (see Section 4). Previous research has demonstrated that every human TCR has some random probability of mimicking any given microbe simply because evolution has highly conserved functional sequences across all cellular life [1,2,8]. Thus, a random set of 325 human TCR sequences derived from 135 healthy individuals was used to derive the baseline occurrence of TCR mimicry of microbial proteins sequences in the UniProtKB database. A total of 25 of the most expanded human TCRs isolated and sequenced from 13 T1DM patients, and 35 of the most expanded TCRs from 2 AM patients were then screened for microbial protein sequence matches in an identical manner. The incidence of healthy TCR sequence matches to some of the most common human pathogens and commensal microbes was then compared with the incidences of T1DM or AM TCR sequences, and the Bonferroni-corrected statistical significance of any differences was determined (see Section 4).

TCRs derived from T1DM patients (henceforth “T1DM TCRs”) displayed a significantly increased rate of incidence for only one of the tested viruses: coxsackievirus type B. Epstein–Barr virus (EBV) displayed an increased incidence as well, though not reaching statistical significance (Figure 1). The incidence of TCR matches with the other 35 virus species was either not significantly different from the expected value derived from the healthy control TCRs or was significantly lower than expected (papilloma-, echo-, and other enteroviruses). This result would seem to confirm the association of coxsackieviruses with T1DM onset and may suggest that EBV sometimes plays a similar triggering role.

The results comparing TCRs derived from AM patients (henceforth “AM TCRs”) also displayed a significantly increased incidence for coxsackievirus (both A and B, in this case), as well as for human herpes virus type 8 (HHV8). Once again, the incidence of papilloma virus matches was significantly decreased, as were the incidences of hepatitis C, adenoviruses, and rotaviruses (Figure 2). These results again appear to confirm the previously reported association of coxsackievirus infections as triggers for AM and suggest that HHV8 may also play such a role.

It is worth noting that while TCR similarities with a wide range of bacteriophage are also commonly observed in these studies, in no case did the incidence of matches involving either T1DM TCRs (Figure 3) or AM TCRs rise above (or even equal) the incidence of matches to the TCRs of healthy controls. These bacteriophage data provide an additional control, suggesting that the association of coxsackieviruses to T1DM TCRs and AM TCRs is significant and that there is a significant shift in TCR usage in T1DM and AM.

TCR similarities to bacterial proteins were also investigated. Unlike virus–TCR matches in which both T1DM and AM TCRs exhibited very similar changes in incidence mainly in coxsackieviruses, the distribution of AM TCR similarities to bacteria differed completely from the distribution of T1DM TCR similarities.

T1DM TCRs displayed a significantly increased rate of incidence over the values expected from healthy control TCRs for only one of the bacteria tested: pathogenic *Clostridia* species (*C. perfringens*, *C. difficile*, *C. clostridioforme*, *C. tetani*, and *C. botulinum*) (Figure 4). This result would seem to confirm a previous report that *Clostridia* are possible triggers for T1DM [98]. As in the case of the viruses, a number of mainly commensal bacteria exhibited significantly lower-than-expected rates of T1DM TCR matches; these included commensal *Clostridia*, *Eubacteria*, *Prevotella*, and *Corynebacteria*. Thus, TCR usage was shifted from that which was displayed by healthy individuals.

Significant increases in AM TCR matches were found in group A *Streptococci* (GAS), viridans *Streptococci*, and total *Streptococci* but not in *Clostridia* or any other bacteria tested (Figure 5). The incidence of AM TCR similarities to a variety of commensal bacteria significantly decreased, but these decreases differed from the T1DM TCR deceases, being found mainly among *Bacteroides*, *Lactobacilli*, and *Bifidobacteria* but once again affecting *Eubacteria* and *Corynebacteria*.

Because previous research suggested that T1DM TCRs often mimic the self-antigens targeted in autoimmune diseases such as T1DM [1,2,7,8], a combination of both BLAST and LALIGN was used to explore possible similarities between T1DM TCRs or AM TCRs and the major self-antigens in these diseases. In the case of T1DM, these self-antigens include insulin, the insulin receptor, glucagon, the glucagon receptor, receptor-type tyrosine-protein phosphatase (PTPRN or PTP-1A), and glutamic acid decarboxylase type 1 (GAD1). T1DM TCRs significantly mimicked all of these self-antigens (Table 1). Similarities were also observed to a number of other potential self-antigens in T1DM, including the zinc transporter SLC39A7, which regulated glycemic control in skeletal muscles [99] and insulin secretion [100]; and an insulin activator, insulin control element (ICE) [101,102]. As a control for comparison with the AM TCRs, T1DM TCRs were also tested for similarities to actin (none were found) and myosin (to which 8 of the 25 T1DM TCRs displayed significant matches; Table 1). Examples of these T1DM TCR similarities to coxsackieviruses, *Clostridia*, and self-antigens are provided in Figure 6 and Figure 7, and the full set of similarities to all of the T1DM TCRs tested here is presented in the Appendix A. The Appendix A also show that many of the T1DM TCRs used in this study mimic other T1DM TCRs identified in other studies. Figure 6 and Figure 7 illustrate, additionally, the fact that coxsackieviruses and *Clostridia* each mimic some of the main self-antigens targeted in T1DM, as has previously been demonstrated experimentally [1,98]. In addition, Figure 8 and Figure 9 demonstrate that *C. perfringens* displays a very large number of similarities to the insulin A and B chains, which are the main targets of autoimmunity in T1DM [103,104]. *C. difficile* was shown to have a similar range of very significant similarities to insulin in a previous study [98] that also established the fact that no other bacterium or virus displayed an equal number or quality of similarities. CV was found to have no significant similarities with insulin (confirmed in [98]) but many significant similarities with the insulin receptor (also reported in [98] with a different set of TCRs).

In total, 80% of the TCRs from the patients used in this study displayed similarities to insulin, the insulin receptor, PTPN1, GAD, glucagon, and/or the glucagon receptor. Among a subset of these TCRs that were previously synthesized and tested for specificity, 35% specifically bound to insulin receptor peptides or insulin [1,2,98].

Notably, some of the similarities identified in Figure 7 have previously been linked to diabetes. Protein 2C of coxsackieviruses is highly conserved across B-type enteroviruses [105,106], as well as some A types [107]; is very similar to GAD65 [105,106]; and binds to a diabetes-associated HLA-DR molecule [105,106,108] activating T cells [109]. All of the main structural proteins of coxsackieviruses, including VP1, VP2, VP3, and VP4, also contain highly conserved regions that activate human T cells [110]. T cells from diabetic patients were particularly responsive to regions of VP2 and VP3 [111,112] (see Figure 7 and Figure 10), but VP1 was also a frequent target of the T1DM T cells [111] and appears among the TCR–coxsackievirus similarities listed in Figure 6 and Figure 10 (see also Appendix A). As noted in Figure 6, Figure 7 and Figure 10, the identified TCR–coxsackievirus similarities are generally very highly conserved across dozens and often hundreds of strains, which include B2, B3, B4, A6, A9, and A16 [113,114].

A similar combination of BLAST and LALIGN studies was performed to determine whether AM TCRs mimicked the main self-antigens involved in AM pathology. This was the case. All the AM TCRs mimicked at least one of the main antigenic targets of AM, including cardiac laminins and collagens, cardiac myosin, actin-binding proteins, and dynein (Table 2 and Figure 11 and Figure 12). The possibility that actin-binding proteins and dynein may be targets in AM has not, apparently, been explored previously, so the extensive similarities are illustrated in Figure 13. A number of cardiac receptors were also identified as being similar to some AM TCRs, including the ryanodine, glutamate, and acetylcholine receptors, but not adrenergic receptors. Only one of the AM TCRs mimicked insulin, glucagon, or their receptors (Table 2). Examples of these similarities are presented in Figure 10 and Figure 11, and the full set is presented for all of the AM TCRs in the Appendix A. The Appendix A also demonstrate that twelve of the AM TCRs used in this study are very similar to TCRs identified from T cells invading muscle during myositis, and nine of the AM TCRs mimic TCRs found in Chagas disease patients. As with the T1DM results, these AM results once again illustrate, incidentally, the previously reported fact that coxsackieviruses and Streptococci each mimic some of the main self-antigens targeted in AM (Figure 13) [73,95].

## 3. Discussion

### 3.1. Summary

To summarize, the TCR beta V-D-J regions expanded during both AM and T1DM mimic coxsackieviruses to a statistically significantly greater degree than the TCR beta V-D-J regions derived from healthy individuals do. The T1DM TCRs displayed significant similarities to no other viruses than coxsackieviruses, and AM TCRs mimicked only human herpes type 8, in addition to coxsackieviruses. The TCRs from AM patients also mimicked *Streptococcal* proteins, particularly from GAS, to a significantly greater degree than TCRs from healthy individuals, while TCRs from T1DM patients mimicked *Clostridia* antigens instead. Once again, these TCRs did not significantly mimic any other bacterial proteins. Thus, coxsackieviruses, S*treptococci*, and *Clostridia* were the only microbes to display significant increases in the distribution of TCRs in AM and T1DM patients compared with TCRs from healthy individuals. The TCR analysis performed here does not provide any evidence for other viruses or bacteria playing a significant role in T1DM or AM initiation. Some significant decreases in the distribution of TCR mimics from AM and T1DM patients compared with healthy patients were also observed, solely among commensal microbes, which are discussed in more detail below.

Increased mimicry between AM TCRs, coxsackieviruses, and *Streptococci* was associated with the mimicry of human heart proteins, including myosin, laminin, actin, and actin-associated proteins such as dynactin; and basement membrane collagens. Previous research had demonstrated that *Streptococcal* antigens mimic myosin and laminins [59,73,95,115], and coxsackievirus antigens mimic mainly actin and collagens [73,95]. However, both coxsackieviruses and *Streptococci* proteins mimic some shared myosin sequences so that there is cross-reactivity between antibodies against coxsackievirus and *Streptococci* [116,117]. Actin-associated proteins such as dynactin were also found to mimic coxsackievirus but not *Streptococci*. In sum, TCR sequences from AM patients mimic coxsackieviruses, *Streptococci*, and the main autoantigenic proteins targeted in the disease.

T1DM TCRs that mimic coxsackieviruses and *Clostridia* also mimicked T1DM target proteins, including insulin, the insulin receptor, glucagon, the glucagon receptor, GAD-1, and PTPRN (PTP-Ia). The results presented here, as well as those of previous research [98], have shown that coxsackieviruses mainly mimic the insulin receptor, while *Clostridia* mainly mimic insulin, GAD-1, and PTPRN. However, anti-coxsackievirus antibodies have been reported to cross-react with GAD-1 as well [118,119]. Thus, TCR sequences from T1DM patients mimic coxsackieviruses, *Clostridia,* and the main autoantigenic proteins targeted in the disease.

Note that the main host antigens that are mimicked by TCRs in each disease fall into complementary pairs. In T1DM, diabetic TCRs mimic insulin (which mimics *Clostridia* antigens) and the insulin receptor (which mimics coxsackievirus antigens). It is possible to infer from the complementarity of insulin for the insulin receptor that some coxsackievirus antigens are complementary to *Clostridia* antigens. Similarly, AM TCRs mimic actin (which mimics coxsackievirus antigens) and myosin (which mimics GAS antigens). Since actin and myosin bind to form actinomyosin, some coxsackievirus antigens are likely to be complementary to some GAS antigens. Likewise, laminin (which mimics GAS antigens) and collagen (which mimics coxsackievirus antigens) bind to form the extracellular matrix, and their complementarity again suggests GAS–coxsackievirus antigen complementarity. Thus, at least some of the targets of the autoimmune process in T1DM and AM are complementary proteins, from which it can be inferred that the triggering antigens are also complementary. Experimental evidence supporting this inference is discussed below.

Notably, although coxsackieviruses were identified as a common trigger for both T1DM and AM, BLAST searching revealed no significant similarities between AM TCRs and T1DM TCRs. Additionally, AM TCRs did not mimic diabetes-associated proteins such as insulin, the insulin receptor, glucagon, or the glucagon receptor. Indeed, a PubMed literature search revealed no studies linking AM to an increased risk of T1DM. These results lead to the conclusion that the TCRs activated in AM differ significantly from the TCRs activated in T1DM even though both sets of TCRs mimic coxsackieviruses. Thus, the presence of *Streptococci* must somehow significantly alter the immune response associated with coxsackieviruses as compared with that elicited when *Clostridia* are present. Some T1DM TCRs did mimic myosin but to a much lesser extent than AM TCRs did, and the T1DM TCRs did not mimic actin or actin-associated proteins. The T1DM TCRs that mimicked myosin might be involved in the increased risk of cardiac autoimmunity that has been reported among diabetics [120,121,122].

In sum, TCR sequences associated with T1DM and AM are distinct from one another even though both sets of sequences significantly mimic coxsackieviruses. These TCRs differ in mimicking *Clostridia* and T1DM-associated antigens in T1DM and *Streptococci* and AM-associated antigens in AM. Furthermore, the proteins that the TCRs mimic in each disease tend to fall into complementary pairs, suggesting that the TCRs and the antigens that elicit them each involve complementary relationships as well.

The rest of this Discussion addresses additional evidence linking coxsackieviruses to both T1DM and AM, *Clostridia* to T1DM, and GAS to AM; the surprising nature of the results reported here in terms of TCR mimicry of these putative autoimmune disease triggers; and the degree to which various autoimmune disease theories, such as molecular mimicry, bystander activation, anti-idiotype theory, and complementary antigen theory, are able to make sense of these results. Finally, the potential importance of the results for elucidating autoimmune disease etiologies and for modeling these in novel types of animal models is addressed.

### 3.2. Further Evidence of Both Clostridia and Coxsackieviruses in T1DM

Evidence for an immune response to coxsackieviruses in T1DM patients is legion and is not further reviewed here (e.g., [15,16,17,18,19,20,21,22]). Only one study has ever looked for evidence of enhanced immunity to *Clostridia* in T1DM, and it found that antibodies recognizing *Clostridia* were present in high titers in most human T1DM sera tested [98]. This result is consistent with *Clostridia* having the highest number of proteins with similarities to insulin of any bacterium or virus currently listed in the UniProtKB database (Figure 7 and Figure 8 and [98]). Furthermore, the observation that TCR sequences expanded in T1DM patients mimic *Clostridia* and coxsackieviruses is consistent with a previous study demonstrating that antibodies against coxsackieviruses cross-react with the insulin receptor, while antibodies against *Clostridia* cross-react with insulin [98].

While no clinical studies have been found that have attempted to isolate enteroviruses such as coxsackieviruses at the same time as *Clostridia* or to measure antibody titers for both, *Clostridia* infections have occasionally been associated with new onset T1DM [123,124], and patients with the most severe coxsackievirus infections have been documented to have significantly increased intestinal *Clostridia* populations compared with patients with mild cases [125,126,127]. Additionally, environmental studies have isolated *C. perfringens* and coxsackieviruses A16, B1, and B5 as fecal contaminants simultaneously from oysters and clams [128] and oyster beds [129], making human co-infections quite possible. Additionally, Root-Bernstein et al. [98] calculated that the probability of randomly contracting overlapping *Clostridia* and enterovirus infections is on the same order of magnitude as the risk of contracting T1DM. The same study demonstrated that some antigens derived from *Clostridia* bind directly to antigens derived from coxsackieviruses suggesting that these antigens are antigenically complementary [98].

More directly relevant to the current study, some of the TCR sequences explored above (e.g., Table 1 and Figure 7) were previously synthesized and tested for their ability to recognize T1DM-related antigens (Table 3). Significant binding by the majority of the tested sequences was observed to insulin, glucagon, or the insulin receptor [130]. Notably, many of these TCR sequences were complementary to each other, behaving like idiotype–anti-idiotype pairs (Table 3), most of these TCRs were also recognized as antigens by antibodies against insulin, glucagon, or the insulin receptor, so that TCRs and antibodies were also acting like idiotype–anti-idiotype pairs. Since there is significant cross-reactivity between insulin antibodies from T1DM patients and GAD-1 [131,132,133], it is likely that some T1DM TCRs recognize GAD-1 and that the coxsackievirus and *Clostridia* proteins that these antigens mimic will be recognized by these TCRs as well. Thus, autoimmune diseases may originate as civil wars involving the immune system attacking itself.

### 3.3. Further Evidence of Both Coxsackieviruses and Streptococci in AM

As with T1DM, the association of coxsackieviruses and AM is very strong and is not reviewed here (e.g., [48,49]). More relevant to the identification of both coxsackieviruses and *Streptococci* by AM TCRs is the evidence of unusually frequent and high antibody titers against both microbes in 65 and 90 percent of AM patients [75,76,77,78,79,80]. Thus, antibodies against both coxsackieviruses and *Streptococci* and the expansion of T cells mimicking both microbes are present in the majority of AM patients who have been tested for both. These complementary results each suggest recent, possibly overlapping or concurrent infections with both microbes. Additional clinical studies have found significantly increased antibody titers against coxsackieviruses in rheumatic heart disease patients infected with *Streptococci* [134], coxsackievirus itself has been isolated from myocardial tissue in Streptococcal-associated RHD [135], and the necessity for both in the etiology of AM/RHD has been posited [136]. Indeed, in animal models, the combined infection results in significantly enhanced myocardial damage compared with either microbe alone [81].

Experimental evidence demonstrates that some coxsackievirus antigens are complementary to *Streptococcal* antigens. Antibodies against coxsackieviruses and, more generally, enteroviruses bind to antibodies against *Streptococcal* antigens with high affinity and specificity resulting in the equivalent of idiotype–anti-idiotype complexes [73,95]. These antibodies also cross-react with myosin, laminin, actin, and/or collagen IV (summarized in Table 4).

TCRs associated with AM have not, at present, been synthesized or tested in experiments equivalent to those summarized for T1DM above. However, AM-derived T cells have been demonstrated to recognize both the M protein of GAS and cardiac myosin [137].

In sum, epidemiological, clinical, and in vitro studies are consistent with the AM TCR data implicating a combination of coxsackievirus–GAS infections as a cause of AM, and this combination helps to explain why AM is so rare, even though both agents are common infections.

### 3.4. Explaining the Results, 1: Molecular Mimicry Theory

So, how can this mimicry between TCRs and the putative triggers of AD be explained? Until recently, the most widely accepted theory of autoimmunity has been the molecular mimicry theory (MMT) [137,138,139], although significant skepticism about the theory’s adequacy has been growing in recent years [140,141,142,143]. MMT originated in Damian’s observation that pathogenic microbes often adapt to mimic their host’s “self”-antigens as a form of molecular camouflage from the host’s immune system [144]. MMT proposes that mimicry between microbes and host proteins might induce an immune response to the microbes that cross-reacts with the host proteins, causing autoimmune disease (Figure 14). MMT can therefore explain the observation reported here (and in [98]) that *Clostridia* mimic insulin and GAD-1, and coxsackieviruses mimic the insulin receptor; or that GAS mimic myosin and laminins [105,117,137], and coxsackieviruses mimic actins and collagens [79,95]. However, MMT does not explain why the infections associated with T1DM and RHD/AM are so common and yet these ADs are very rare. More importantly, mimicry of TCRs for microbes associated with triggering autoimmune diseases must be surprising since TCRs are supposed to be complementary, not similar, to foreign antigens. There is nothing in molecular mimicry theory to suggest or predict that TCRs should, or even could, mimic the microbial antigens associated with triggering disease.

### 3.5. Explaining the Results, 2: Bystander Activation

MMT has a major flaw in that it has proven to be impossible to induce autoimmune disease in animal models with molecular mimics in the absence of adjuvants. Bystander activation [145,146,147,148] has been proposed as a “fix” for MMT that explains the role of adjuvants, as well as the observation that people who develop autoimmune diseases are often infected with more than one microbe and display hyperactivated innate immunity (see Introduction). The theory proposes that preceding or concurrent infections unrelated to the autoimmune disease may provide the necessary innate activation required to break tolerance to “self”-mimicking epitopes. Combined infections may indeed be required to initiate and sustain the hyperactivation of innate immunity necessary to support autoimmune disease pathology by stimulating sets of synergistic Toll-like receptors and nucleotide-binding oligomerization domain (NOD)-like receptors (NLRs) [149,150,151]. However, bystander activation cannot explain the TCR data reported here and similar data provided in our other studies [1,2,8,73,89,95,96,97,98,130], which demonstrate that each autoimmune disease is characterized by a unique microbial pair. This would not be the case if the bystander infection were providing nothing more than a non-specific adjuvant effect since many co-infections should provide the necessary innate activation. The crucial TCR result to be explained here is the identification of the specific combination of coxsackieviruses with *Streptococci* in AM and the specific combination of coxsackieviruses with *Clostridia* in T1DM. Nonspecific bystander activation, or an adjuvant-like effect, cannot explain these observations. However, bystander activation does have one clear advantage over MMT alone, which is that the necessity for a secondary infection goes a long way toward explaining why ADs are rare even when their infectious triggers are common: combined or overlapping infections are much less likely to occur in individual patients than are single infections.

### 3.6. Explaining the Results, 3: The Anti-Idiotype Theory

Several additional explanations for the TCR–antigen mimicry results are possible that build on various aspects of MMT and bystander activation and address their limitations with regard to the data produced here. The first derives from Plotz’s anti-idiotype theory of AD [152,153] in combination with molecular mimicry. Assume that microbe 1 expresses a dominant antigen that mimics some human protein 1. The antigen will expand an idiotypic TCR that will also recognize human protein 1. Assume further that the microbial antigen is complementary to a second human protein (protein 2) that it uses as a cellular receptor for targeting its infection. If the idiotypic TCRs are expressed at high enough levels for enough time, they may elicit an anti-idiotypic set of TCRs. These anti-idiotypic TCRs will be complementary not only to the idiotypic TCRs but also to human protein 2 (the microbial receptor). The result will be that the idiotypic and anti-idiotypic TCRs will be complementary, neutralizing each other; the idiotypic TCRs will mimic protein 2 (the microbial receptor); the anti-idiotypic TCRs will mimic both the microbial antigen and human protein 1. Logic dictates that human proteins 1 and 2 will also be complementary to each other (Figure 15).

This theory provides a basis for explaining the existence of two sets of TCRs in AD and predicts that they will be complementary to each other (i.e., acting like idiotype–anti-idiotype pairs). No evidence for the presence of anti-idiotypic TCRs in T1DM or AM appears to exist, but indirect evidence comes from the fact that the TCRs mimic both actin and myosin (which combine to form actinomyosin), as well as collagen IV and laminins (which combine to form the extracellular matrix; Table 2). Similar evidence for TCR complementarity in T1DM and AM exists in the observation that T1DM TCRs mimic both insulin and its receptor (Table 1 and Table 3). However, the anti-idiotype theory leaves unexplained why the complementary TCRs mimic two distinct and very specific sets of microbial antigens: the theory requires only one microbial trigger. So, why do combinations of coxsackieviruses and *Streptococci* occur in AM while combinations of coxsackieviruses and *Clostridia* occur in T1DM? The anti-idiotype theory also has no explanation for why the vast majority of people who contract an infection such as coxsackievirus does not develop T1DM or AM since only a single infectious trigger is required according to the theory: as with MMT, the anti-idiotype theory does not explain the rarity of AD.

The anti-idiotype theory of AD, in short, can produce the kind of TCR data described here but provides no role for a second microbe in disease pathology and therefore leaves unexplained how coxsackieviruses can induce two distinct diseases.

### 3.7. Explaining the Results, 4: Complementary Antigen Theory

A third possible explanation of the TCR mimicry of foreign and self-antigens in autoimmune diseases is provided by the complementary antigen theory of AD [154,155,156,157,158,159,160,161,162]. According to this theory, ADs are triggered by a pair of molecularly complementary antigens contributed by a pair of concurrent, overlapping or temporally related infections. Each infection is characterized by antigens that mimic some set of human “self”-antigens, and each infection elicits a set of idiotypic TCRs to its dominant antigens. If some of these antigens are complementary, some of the resulting TCRs will also be complementary and behave like idiotype–anti-idiotype pairs. Because the microbial antigens are complementary and mimic human “self”-antigens, the “self”-antigens will be complementary to each other, and they will also become targets of the TCR elicited by the microbial antigens (Figure 16). Furthermore, since the TCRs are complementary to each other, the “self”-antigens are complementary to each other, and the TCRs are complementary to the “self”-antigens, each TCR will mimic one of the sets of “self”-antigens. The result will be an inability of the immune system to differentiate between “self” and “non-self”, providing a mechanism for breaking immune tolerance: The immune system has a “choice” of either responding to both sets of complementary microbial antigens, thereby inducing a “civil war” within itself, or tolerizing itself against both sets of microbes, thereby abrogating its ability to address the infections effectively. Finally, the complementary antigen theory, like bystander activation, explains why autoimmune diseases are rare: the probability of developing overlapping or concurrent infections such as coxsackievirus with *Streptococci* or *Clostridia* is very small.

Most importantly, complementary antigen theory explains how coxsackieviruses can be associated with multiple ADs but result in different pathologies depending on the bacterial co-infection with which they are associated. As Table 2 summarizes, the TCRs expanded in AM always mimic some “self”-protein targeted by autoimmunity in AM, such as myosin, laminin, or actin-associated proteins, while these expanded TCRs very rarely mimic T1DM targets such as insulin or the insulin receptor. Conversely, TCRs expanded in T1DM all mimic some set of “self”-proteins targeted in T1DM but rarely mimic “self”-proteins associated with AM. Thus, although coxsackieviruses are implicated in both diseases, the TCR expanded in each disease is distinct, as would be expected if different sets of complementary antigens were present in GAS as compared with *Clostridia*. Additionally, studies of antigen processing in macrophages and monocytes have demonstrated that complementary antigens are processed differently than the individual components, providing a mechanism both for avoiding tolerance and for activating different TCRs against different sets of coxsackievirus antigens depending on the co-infection (reviewed in [163]). Finally, complementary antigen theory also explains the presence of experimentally demonstrated complementary TCRs in T1DM (Table 3) and predicts that such complementary TCRs will also characterize those expanded in AM.

To be clear, this theory posits that TCRs mimic the inducing antigens of an autoimmune disease only because those antigens are complementary to each other. Thus, the TCRs mimicking each antigen are, in fact, elicited by the complementary antigen. This antigen complementarity has been directly demonstrated in the case of T1DM for some coxsackievirus and *Clostridia* antigens [1]; and indirectly demonstrated for T1DM by the demonstration that coxsackievirus antibodies bind with high affinity and specificity to some *Clostridia* antibodies [1], and for AM by the high affinity and specificity for coxsackievirus antibodies for some *Streptococcal* antibodies [73,95]. Demonstrating antigen complementarity is therefore a crucial prediction that differentiates this theory from the bystander, MMT, and anti-idiotype theories.

### 3.8. Explaining the Results, 5: Antigen Templating of the Hypervariable Region

A fourth possible explanation of the TCR data presented here is the most speculative and antidogmatic but is fully compatible with complementary antigen theory and would also explain the observations summarized in the Introduction that TCRs expanded in HIV infections mimic HIV antigens [3,4,7] and TCRs expanded in SARS-CoV-2 infections mimic coronavirus antigens [8]. The phenomenon of immune system molecules mimicking microbial antigens has been observed previously in the case of the CRISPR-Cas system of immunity employed by bacteria to protect themselves against bacteriophage. One of the observations that led to the discovery of this system was that bacteria that resisted bacteriophage infections had copies of bacteriophage exons introduced into their genomes [164]. The purpose of these bacteriophage sequences is to produce antisense RNA to block the replication and maturation of bacteriophage if encountered again. One can speculate that the vertebrate immune system evolved to respond to infections in a similar way (Figure 17). In essence, such a mechanism might employ the RAG-1 and RAG-2 enzymes, which are notably related to viral and bacterial integrases [165,166], to insert antigen-associated genomic material (or reverse-transcribed RNA) into the hypervariable region of the genes encoding TCR, B-cell receptors, and antibodies, thereby initiating the expansion of the clone. A variety of groups have, over many decades, suggested that such a mechanism might exist and provided arguments for why such a mechanism might be required in place of the textbook mechanism of random clonal production of every possible TCR or antibody sequence (e.g., [167,168,169,170,171,172,173]. This is not the place to provide a detailed description of how such a mechanism might work or the data and logical arguments that might support such a mechanism (some of which are provided in [174,175]). However, in the spirit of examining all possible hypotheses that can explain TCR mimicry of AD-inducing antigens, antigen templating of the hypervariable region is compatible with the results reported here.

### 3.9. Experimental Tests to Differentiate the Various Theories

Each of the theories outlined above makes predictions that are distinct from the others, providing a means of evaluating their relative explanatory value above and beyond their ability to explain already existing data. The molecular mimicry theory predicts that T1DM and AM should both be induced by coxsackieviruses, which is clearly not the case. An alternative proposed here is that the involvement of *Clostridia* in T1DM differentiates its induction from that of AM. To test this prediction, one necessary test would be to determine whether *Clostridia* or particular *Clostridia* antigens identified above can induce T1DM but not AM. The particular hypothesis proposed here—that both diseases require coxsackievirus involvement—would predict that *Clostridia*, alone, will not suffice to induce T1DM. Nonetheless, this possibility must be tested as a control for the additional tests below.

The equivalent predictions regarding AM have already been tested. Previous investigators have established that coxsackieviruses by themselves (that is, in the absence of cardiac proteins) cannot induce AM, and neither can *Streptococci* or *Streptococcal* antigens, which require an adjuvant such as Freund’s complete adjuvant. Therefore, the molecular mimicry theory cannot explain AM/RHD etiology. What has not been tested thus far is whether the induction of an anti-idiotypic response is capable of initiating AM. Can repeated injections of coxsackieviruses or coxsackievirus antigens, without cardiac proteins or any adjuvant, induce sufficient immunity to produce anti-idiotypic antibodies and/or T cells, and if so, does the induction of anti-idiotypes induce the characteristic pathology? Such tests of the anti-idiotype theory are also required for *Clostridia* with regard to T1DM and with *Streptococci* for AM. Again, the theory proposed here predicts that these tests will not result in autoimmune disease.

The primary test of the complementary antigen theory is to infect appropriate animals with both coxsackievirus and either *Clostridia* or *Streptococci*. Alternatively, since both T1DM and AM are autoimmune diseases, it should be possible to induce either disease using killed coxsackievirus (or coxsackievirus antigens) combined with either inactivated *Clostridia* (or appropriate *Clostridia* antigens as predicted above) or *Streptococci* (or appropriate *Streptococcal* antigens). In light of the fact that the ratio of adjuvant to antigen in most autoimmune disease models must be kept within certain bounds, it is possible that infectious doses and the timing of the infections relative to each other will need to be optimized.

Additional tests of the complementary antigen theory are also possible. One is to induce antibodies and/or T cells in an inbred strain of rodents. One set of animals can be immunized against coxsackievirus, and the other against either *Clostridia* or *Streptococci*. The antibodies and/or T cells from each set of animals can be isolated and passively transferred to previously uninoculated animals. Animals receiving only coxsackievirus antibodies/T cells or only the bacterial antibodies/T cells are predicted to develop no autoimmune pathologies. However, animals receiving a combination of antibodies/T cells from both coxsackievirus-immunized and bacteria-immunized animals are predicted to develop either T1DM or AM depending on the bacterium. Since polyclonal antibodies against coxsackieviruses, Streptococci, and Clostridia are available commercially, a shortcut to performing the same type of experiment might be to combine, say, rabbit anti-coxsackievirus antibodies with rabbit antibacterial antibodies and inoculate rabbits with the combinations. In all of these experiments, a further control would be to use antibodies/T cells derived from animals inoculated with bacteria predicted by this study to be unrelated to T1DM or AM etiologies. Similarly, the use of coxsackievirus antibodies/T cells can be controlled for by using antibodies/T cells derived from immunization against viruses predicted by this study to be unrelated to T1DM or AM.

In vitro studies can also test some of the predictions made by the hypothesis proposed here. Some of these are referred to in the Results section and Discussion above. For example, the complementarity between particular coxsackievirus antigens and those derived from *Streptococci* or *Clostridia* can be tested using enzyme-linked immunoadsorption assays and other immunological techniques, as can the cross-reactivity of antibodies against these antigens for the relevant self-proteins. T-cell activation studies can be performed to determine whether clones respond similarly to both microbial antigens and their self-protein mimics predicted here. Additionally, such T-cell activation studies can be modified to permit tests of whether T cells activated against coxsackievirus antibodies behave like anti-idiotypic T cells or T cells activated against either *Clostridia* or *Streptococci*.

Clinical studies can be performed to determine how frequently coxsackievirus infections coincide temporally (either immediately following, concomitant with, or complicating) *Clostridia* and *Streptococcal* infections. Clinical studies can also be performed to address whether immunity to coxsackievirus and *Clostridia* antigens occurs in T1DM and, correspondingly, to coxsackieviruses and *Streptococci* in AM. The hypothesis presented here predicts that titers of antibodies representative of recent infections will be present against both microbes in each disease and, correspondingly, that T-cell clones against both microbes will similarly be unusually expanded, indicating recent infection. The hypothesis does not predict that both infections will be present in their active forms because the autoimmune disease state is not due to the infections themselves but to their resulting immune responses.

Finally, the testing of antigen templating of the hypervariable regions of antibodies and T-cell receptors needs to be tested. Since this possibility is seriously antidogmatic, the number and types of tests required go far beyond the space permitted in the present paper. I will only point to two articles that have pioneered some of the appropriate techniques and demonstrated their clinical utility [176,177].

### 3.10. Elucidating AD Etiologies from TCR Sequences to Develop Novel Animal Models

The most important implications of this research are the possibility that autoimmune disease etiologies can be elucidated from the TCR sequences expanded preferentially in each disease and that these etiologies can then be tested through novel animal models. Given large enough TCR databases for setting normal sequence similarity distributions, it might even be possible to utilize TCR sequences from individual patients to identify their unique microbial triggers.

New animal models might permit breakthroughs not only in understanding etiologies but in the development of novel preventative vaccines and disease therapies. Previous attempts to induce T1DM with coxsackieviruses have universally failed (see Introduction), but no one has yet attempted to induce the disease in any animal by using coxsackieviruses in combination with bacteria such as *Clostridia.* Two existing animal models are suggestive. The first involves initiating T1DM in Lewis rats by infecting them with Kilham rat virus (KRV). KRV increased intestinal *Bifidobacterium* and *Clostridium* species [178] in parallel with T1DM development such that antibiotic treatment not only prevented the increase in *Bifidobacterium* and *Clostridium* abundance but also prevented the onset of T1DM [175]. Similarly, the susceptibility of NOD mice to the development of T1DM depends on gut *Clostridial* butyrate biosynthesis species [179]. Probiotic *Clostridium butyricum* [180,181] or fecal transplants with commensal *Clostridia* [182] were found to protect the mice against T1DM. Conversely, vancomycin-treated NOD mice, which experienced a significant decrease gut *Clostridia* species, were much more prone to develop T1DM than NOD mice not treated with antibiotics; the accelerated T1DM risk was correlated with an increase in pathogenic forms [183]. A T1DM animal model that employs both enteroviruses and *Clostridia* might therefore permit novel insights into human diabetes that artificial models such as streptozotocin-treated and spontaneous models such as NOD and BB/EE mice cannot.

Similar benefits might follow from novel AM models since current models fail adequately to mimic the etiology and pathogenesis of the human disease [184]. One typical animal model involves the use of inactivated GAS, group G *Streptococci*, or their M proteins repeatedly inoculated subcutaneously in Freund’s complete adjuvant (FCA: inactivated *M. tuberculosis* in mineral oils) stimulated with additional *B. pertussis* intraperitoneal inoculations [61,71]. The FCA cannot be omitted or replaced with nonmicrobial adjuvants [70]. On the one hand, obviously, no human being contracts autoimmune heart disease in this manner. On the other hand, the requirement of FCA and *B. pertussis* to supplement the GAS or GGS in order to initiate autoimmunity strongly points to the probability that human AM similarly requires multiple microbial infections. As noted above, epidemiological and clinical studies suggest that a vast majority of AM patients are co-infected with coxsackieviruses and *Streptococci*, and Kogut [81] reported much more extensive cardiac damage with the virus–bacterium combination than with either microbe alone. Exploring whether coxsackieviruses can replace FCA–pertussis in GAS animal models is therefore warranted. Indeed, live coxsackievirus without FCA or pertussis is often used to induce AM in mice, but notably the coxsackievirus must passage through cardiac tissue first; some of that cardiac tissue is co-inoculated into the mice, along with the virus [65,172]. Since it is well known that *Streptococci* mimic cardiac antigens such as myosin and laminin [59,185], it would again be logical in light of the TCR data reported here to attempt to induce AM in mice or rats by using coxsackievirus that has not been heart passaged, in combination with *Streptococci*.

The approach to identifying possible triggers of autoimmune diseases by using TCR mimicry of microbial antigens that was used here is widely extendable. A previous study observed that Crohn’s disease (CD) TCRs had no similarities to T1DM TCRs in their microbial similarity profiles. CD TCRs very significantly mimicked *Cryptococcus neoformans* and *Pseudomonas aeruginosa* and, to a lesser extent, *Corynebacteria* and *Enterobacteria*, but no other bacteria and no viruses [2]. Notably, each of these TCR-identified bacteria has individually been associated clinically as possible triggers of CD (reviewed in [2]), but combinations of them have never been explored as synergistic triggers. Again, such experiments are warranted by the TCR data and could produce a novel and more human-like model for understanding, preventing, and treating CD.

Additionally, the same approach employed here identified combinations of SARS-CoV-2 with Streptococci as a probable inducer of COVID-19-associated myocarditis and *Enterococcus faecium*, and *Staphylococcal* antigens in the Kawasaki-like multisystem inflammatory syndrome in children (MIS-C) [8,185].

Other autoimmune disease etiologies may follow, and additional animal models, preventative approaches, and treatments may follow with them.

### 3.11. Implications for Loss of Microbiome Tolerance in Autoimmune Diseases

Yet another implication of the results reported here is that they may elucidate the specific microbiome alterations that accompany every autoimmune disease. I have previously suggested that because the microbiome evolves to mimic host TCRs, autoimmunity will attack not only host antigens but also similar antigens expressed by microbiome constituents [1]. Figure 1, Figure 2, Figure 3, Figure 4 and Figure 5 illustrate the fact that TCRs mimicking mainly commensal microbes become less prevalent among T1DM and AM patients. The decrease in TCRs mimicking these commensal microbes seems to correlate with significant increases in the prevalence of these microbes. For example, *Bacteroides* and *Lactobacillus* species increase in T1DM patients [186,187,188], and TCRs mimicking these species decrease (Figure 4). *Bacteroides* and *Firmicutes* are similarly elevated in AM [189,190], which correlates with a significant decrease in the TCR mimicry of these microbial species. Notably, enteroviruses are well known to exploit direct interactions with commensal microbiota to enhance and disseminate infection [191,192,193,194,195,196,197] so that antibiotic treatment of animal susceptible to experimental AM can prevent coxsackievirus-induced disease [198], as can fecal transplants correcting dysbiosis [191]. Fecal transplants have also been reported to be effective for treating T1DM [199,200]. If TCR sequence distributions accurately reflect changes in microbiota, then they may provide a useful tool for tailoring specific probiotic treatments for individual patients. This possibility clearly requires a great deal of further investigation, however.

### 3.12. Implications for Coxsackievirus and Clostridia Vaccine Development

One final implication of this study relates to the development of vaccines against enteroviruses, Streptococci, and Clostridia, both for purposes of infection prevention and also for the specific prevention of diabetes and myocarditis. Substantial research is being devoted to developing coxsackievirus vaccines [201,202,203,204], some specifically for the prevention of diabetes [205,206,207,208,209], and some specifically for the prevention of myocarditis [210,211,212,213,214]. Extensive research is also ongoing toward the development of *Clostridia* vaccines [215,216,217,218,219,220]; however, the connection of *Clostridia* to diabetes is too recent to have resulted in any of these vaccines being purposed specifically toward the prevention of diabetes. A wide range of pneumococcal vaccines already exist, but ongoing efforts are underway to develop *S. pyogenes* and other Group A Streptococcal vaccines, particularly for the prevention of myocardial disease sequelae in high-incidence areas of the world [221,222,223]. What none of these development programs has done, as far as I have been able to determine, is utilize host–microbe protein similarity data to antigen-delete or (in this age of mRNA vaccination) to modify the vaccine sequences to reduce or eliminate high-similarity regions that might, under some circumstances, increase the risk of diabetes or myocarditis.

### 3.13. Limitations of the Study

This study has limitations. The most important implications have not yet been tested. These include testing the predictions made above by devising new animal models of AM and T1DM, employing combined infections (or mixtures of killed or synthetic antigen mixtures); direct tests of antigen complementarity, employing physicochemical techniques to demonstrate direct binding; clinical demonstrations of *Clostridium* infections in T1DM patients coinciding temporally with, or complicating, coxsackievirus or other enterovirus infections; similar clinical demonstrations of *Streptococcal* infections coinciding temporally with, or complicating, coxsackie infections or other enterovirus infections preceding AM; and exploring whether changes in TCR distributions directly identify corresponding microbiome alterations associated with each disease.

This study is also limited by being based on a limited number of T1DM and AM patients’ TCRs. Thus, while the similarities to enteroviruses/*Clostridia* in T1DM and enteroviruses/*Streptococci* in AM are very strong, such correlations do not prove causation, nor do they preclude the possibility that other virus–bacterium combinations discussed in the Introduction may trigger these autoimmune diseases. The best that can be concluded is that other combinations are much less likely. However, only with larger sets of TCRs will such exceptions become observable using the current approach.

Another limitation is the use of small sets of TCR sequences that were highly expanded in individual patients. While these TCR sequences were derived from the most highly activated T cells in the patients, it is presently unknown what the optimal range or number of sequences should be analyzed from any individual patient or group of patients in order to best identify possible autoimmune disease triggers. Analyzing too few TCR sequences may miss critical microbial similarities, while analyzing too many may swamp out the identification of possible triggers amidst a plethora of irrelevant data. The best strategy may be to increase the number of individual sets of TCR sequences used rather than the number of TCR sequences derived from each individual. Such expanded sets of data would undoubtedly resolve whether some of the not-quite-statistically significant observations observed here are “real” or not.

A fourth limitation of this study is that the TCR sequence similarity analysis was performed “by hand”. The difficulty in entering and curating each result individually necessarily limited the number of sequences that could be handled in a reasonable amount of time. We are currently working on an automated computer program to handle TCR–microbial similarity searches more efficiently and extensively. This program may also permit larger numbers of controls, thus improving the statistical identification of outliers or exceptions to the coxsackievirus/*Streptococcus* and coxsackievirus/*Clostridium* correlations reported here.

Finally, the results reported here might be entirely artifactual due to contamination of the TCR sequences by viral or bacterial sequences. This possibility is unlikely, however. The TCR sequences utilized in this study (see sources listed in Section 4 below) were identified DNA primers designed to recognize highly conserved, genetically encoded TCR sequences immediately preceding the V-D-J regions that were sequenced. Thus, the viruses and bacteria that are overrepresented in our analysis would have had to have mimicked not just the variable region of the TCR but also the region preceding it. While theoretically possible, such highly conserved identities were not found in this study, and they are not reported elsewhere.

In short, this is a pioneering effort with all of the limitations that initial explorations inevitably have, and subsequent studies will undoubtedly find ways to conduct the type of analysis trialed here, using better methods that either validate or invalidate the results. Its strength is to make a very large number of unique predictions that are clearly amenable to testing in a wide variety of ways.

## 4. Materials and Methods

### 4.1. Similarity Searches

Similarity searches comparing TCR sequences with virus, bacteria, and human proteins were carried out using the standard protein BLASTP (protein–protein similarities) at the National Center for Biotechnology Information (NCBI) at the National Library of Congress, Washington, DC, USA. (https://blast.ncbi.nlm.nih.gov/Blast.cgi?PAGE=Proteins, accessed between 1 January 2023 and 1 November 2023). Each TCR sequence was input as a FASTA sequence; the UniProtKB/SwissProt database was selected with an appropriate organism limitation (viruses, taxid 10239; bacteria, taxid 2; *Homo sapiens*, taxid 9606). The general parameters were set with 1000 sequences to display; E = 100; threshold at 0.5; initiating word size, 2; BLOSUM80; and filtering for low-complexity regions. The human matches were limited to E < 101 after the search was completed so as to ensure high-quality matches and curated to eliminate matches to other TCR sequences. The resulting matches were hand-curated to identify the approximately 40 viruses, bacteria, and human proteins analyzed in the figures presented in this study. The selection of these particular viruses and bacteria was based on a previous study [1] in which their similarity profiles were evaluated in terms of type 1 diabetes and Crohn’s disease. The human proteins chosen for analysis were chosen in terms of their likelihood of being involved in myocardial or diabetic pathologies.

Having identified *Clostridia*, *Streptococci*, and enteroviruses as having statistically significant (see below) similarities to AM and T1DM TCRs in the general virus and bacteria searches, further in-depth BLASTp searches were conducted, as above, but comparing the TCRs with *Clostridia* (taxid:186801), *Clostridiales* (taxid:186802), *Streptococcus* (taxid:1301), and Enterovirus (taxid:12059).

LALIGN similarity searches were also carried out to investigate each TCR’s mimicry of human autoantigens associated with T1DM (AM similarities were already carried out in [95]: >sp|P06213|INSR_HUMAN insulin receptor OS = Homo sapiens OX = 9606; >sp|P01308|INS_HUMAN insulin OS = Homo sapiens OX = 9606; >sp|P01275|GLUC_HUMAN pro-glucagon OS = Homo sapiens OX = 9606; >sp|P47871|GLR_HUMAN glucagon receptor OS = Homo sapiens OX = 9606l >sp|Q99259|DCE1_HUMAN glutamate decarboxylase 1 OS = Homo sapiens OX = 9606; >sp|Q16849|PTPRN_HUMAN receptor-type tyrosine-protein phosphatase-like N OS = Homo sapiens.

### 4.2. TCR Sources

Normal TCR sources: 104 sequences from [1] and 221 randomly selected entries from [224,225,226,227,228,229] for a total of 325 sequences derived from 135 individuals.

T1DM TCR sources: [230,231] and selections of T1DM dominant TCR from ([232], Table 2) for a total of 25 TCR derived from 13 patients.

AM TCR source: [233] for a total of 35 TCR from 2 patients.

### 4.3. Statistics

A chi-squared test (http://www.quantpsy.org/chisq/chisq.htm; accessed 4 October 2023) was used to make pair-wise comparisons between the percentage of matches for TCRs to the set of human viruses, bacteria, and proteins selected for analysis (see above). Because multiple chi-squared tests were run for each TCR group, a Bonferroni correction was applied to the resulting *p*-values (http://www.winsteps.com/winman/bonferroni.htm; accessed 6 October 2023). Because no significant difference was demonstrated between the percentage or overall distribution of the healthy TCR group as compared with randomized TCR sequences and antisense TCR sequences in a previous study [1], it was assumed that TCR sequences from random healthy individuals could be used as a control for statistical purposes in this study.

## Figures and Tables

**Figure 1 ijms-25-01797-f001:**
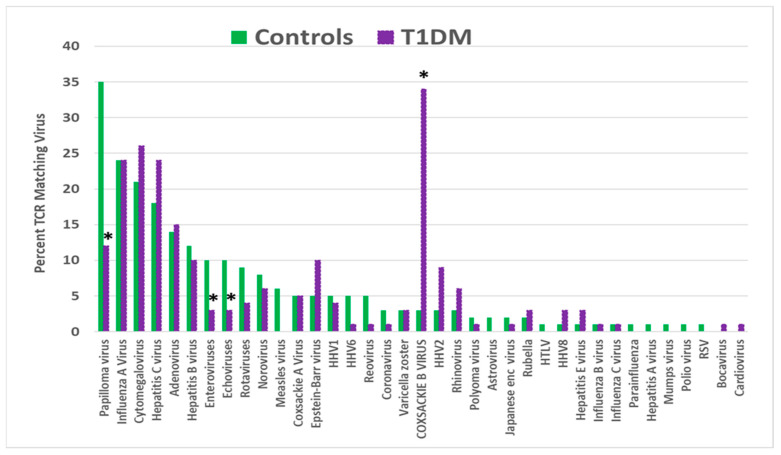
Percentage of T-cell receptor (TCR) sequences from type 1 diabetes mellitus (T1DM) patients that mimic 37 common human viruses compared with the percentage mimicked by TCRs from healthy individuals. Asterisks (*) indicate that the difference is statistically significant after Bonferroni correction (see Appendix B, Table A1). HHV = human herpes virus; HTLV = human T-lymphotropic virus; RSV = respiratory syncytial virus.

**Figure 2 ijms-25-01797-f002:**
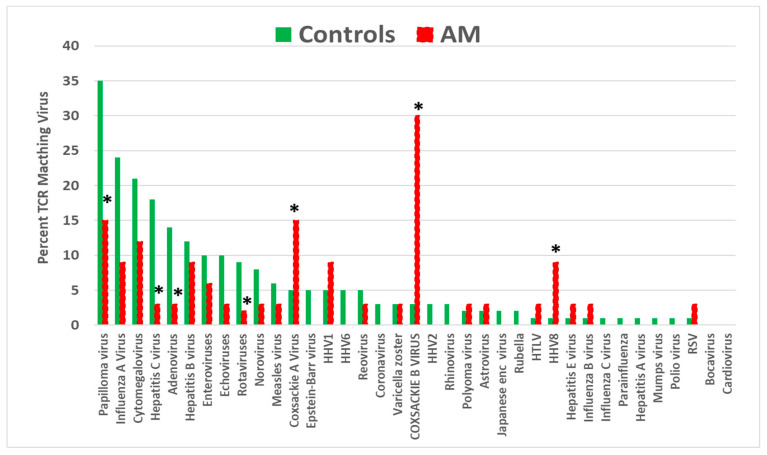
Percentage of T-cell receptor (TCR) sequences from autoimmune myocarditis (AM) patients that mimic 37 common human viruses compared with the percentage mimicked by TCRs from healthy individuals. Asterisks (*) indicate that the difference is statistically significant after Bonferroni correction (see Appendix B, Table A1). HHV = human herpes virus; HTLV = human T-lymphotropic virus; RSV = respiratory syncytial virus.

**Figure 3 ijms-25-01797-f003:**
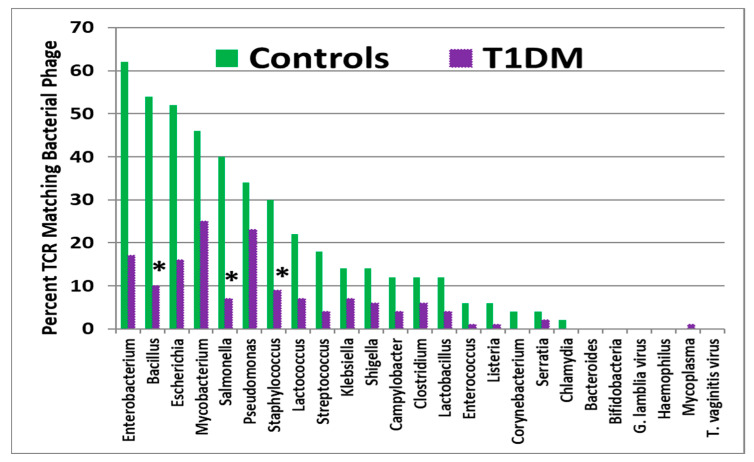
Percentage of T-cell receptor (TCR) sequences from type 1 diabetes mellitus (T1DM) patients that mimic 25 bacteriophages compared with the percent mimicked by TCRs from healthy individuals. Asterisks (*) indicate that the difference is statistically significant after Bonferroni correction (see Appendix B, Table A2).

**Figure 4 ijms-25-01797-f004:**
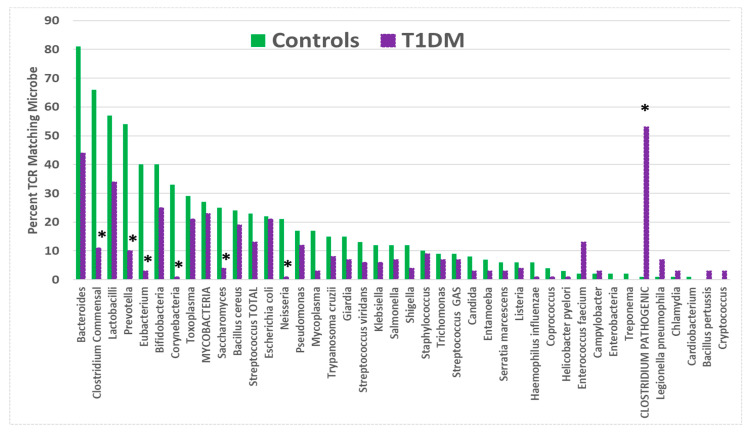
Percentage of T-cell receptor (TCR) sequences from type 1 diabetes mellitus (T1DM) patients that mimic 42 common human bacteria compared with the percentage mimicked by TCRs from healthy individuals. Asterisks (*) indicate that the difference is statistically significant after Bonferroni correction (see Appendix B, Table A3).

**Figure 5 ijms-25-01797-f005:**
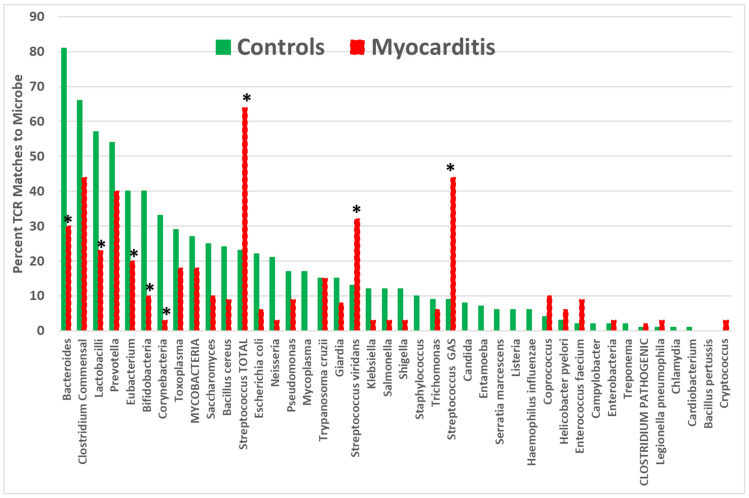
Percentage of T-cell receptor (TCR) sequences from autoimmune myocarditis patients that mimic 42 common human bacteria compared with the percentage mimicked by TCRs from healthy individuals. Asterisks (*) indicate that the difference is statistically significant after Bonferroni correction (see Appendix B, Table A3).

**Figure 6 ijms-25-01797-f006:**
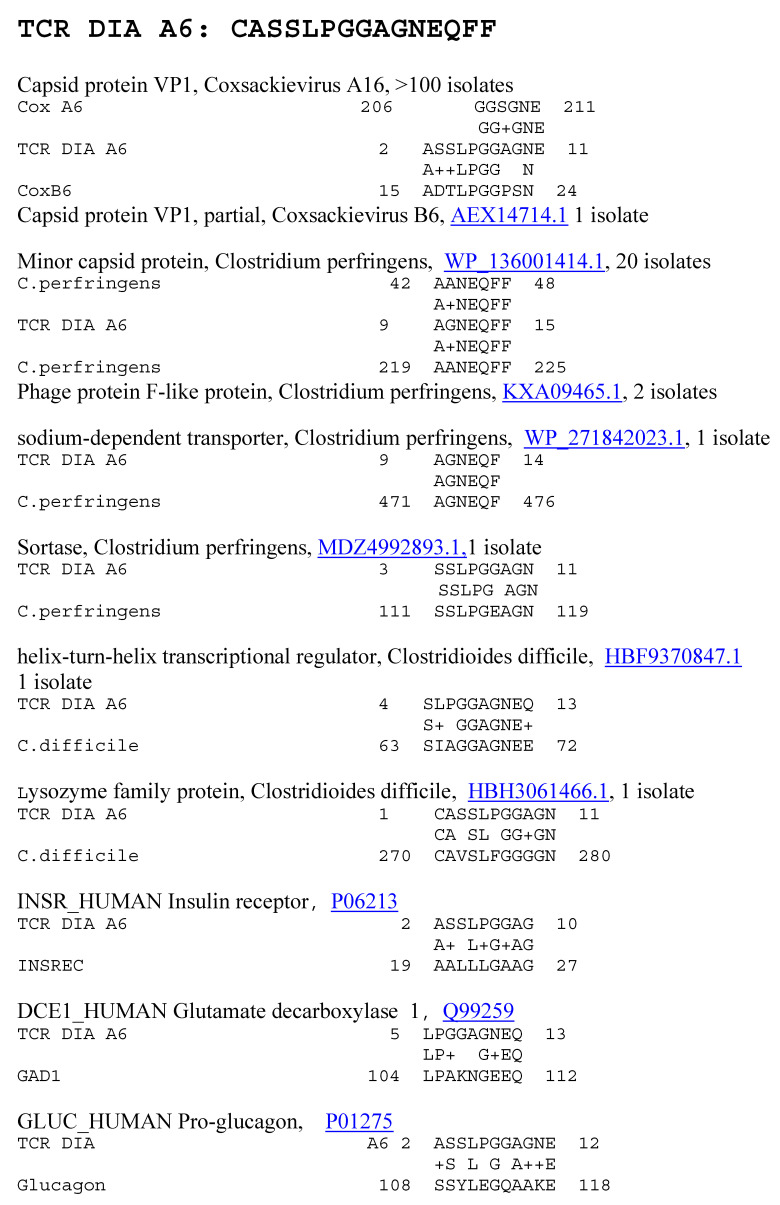
Significant similarities between the TCR DIA A6 sequence and the antigens listed in Table 1 above. Underlined numbers are the UniProtKB identifiers.

**Figure 7 ijms-25-01797-f007:**
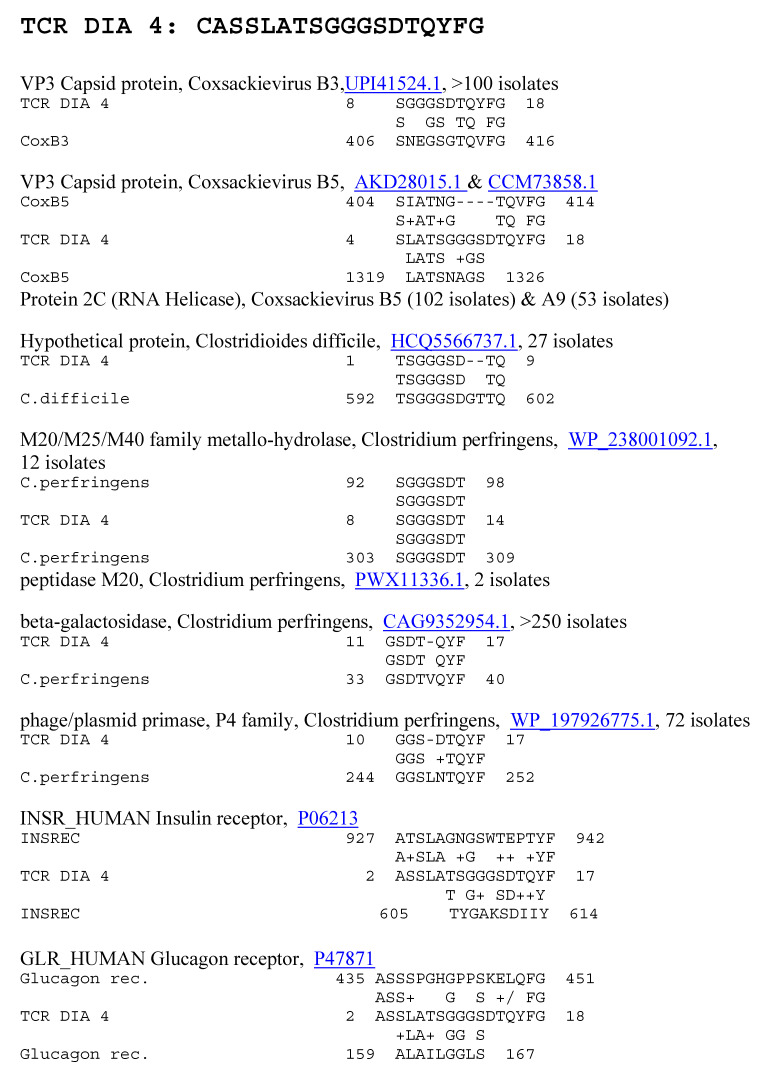
Significant similarities between the TCR DIA 4 sequence and the antigens listed in Table 1 above. Underlined numbers are the UniProtKB identifiers.

**Figure 8 ijms-25-01797-f008:**
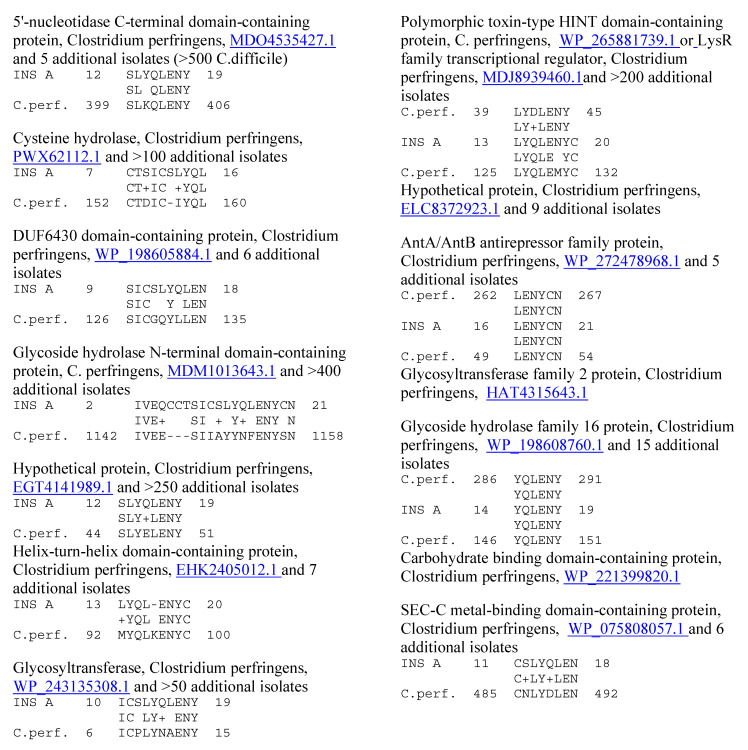
Similarities between Clostridium perfringens (C.perf.) proteins (UniProtKB identifiers are underlined) and the human insulin A chain (INS A), using LALIGN.

**Figure 9 ijms-25-01797-f009:**
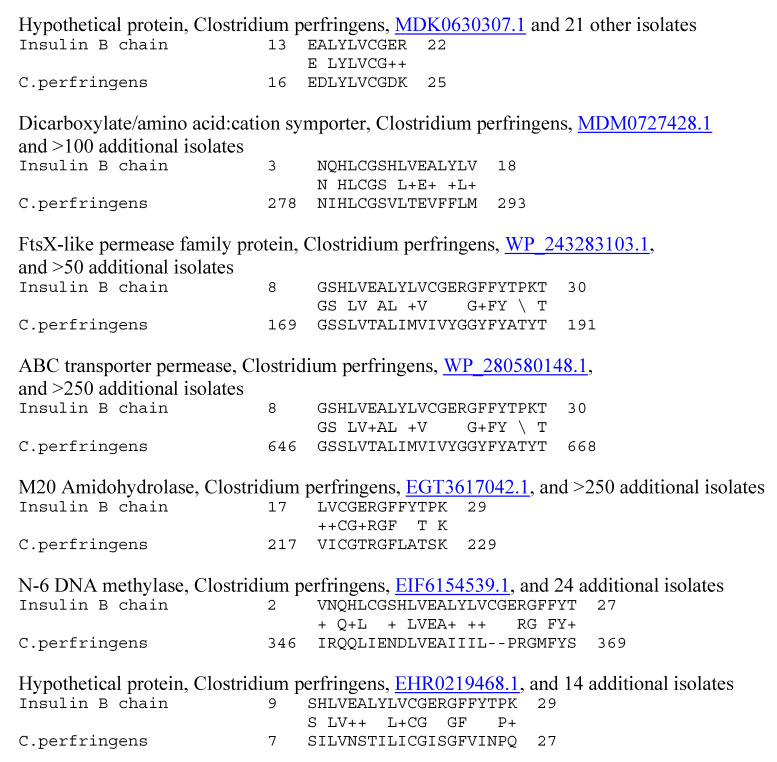
Similarities between Clostridium perfringens (C.perf.) proteins (UniProtKB identifiers are underlined) and the human insulin B chain (INS B), using LALIGN.

**Figure 10 ijms-25-01797-f010:**
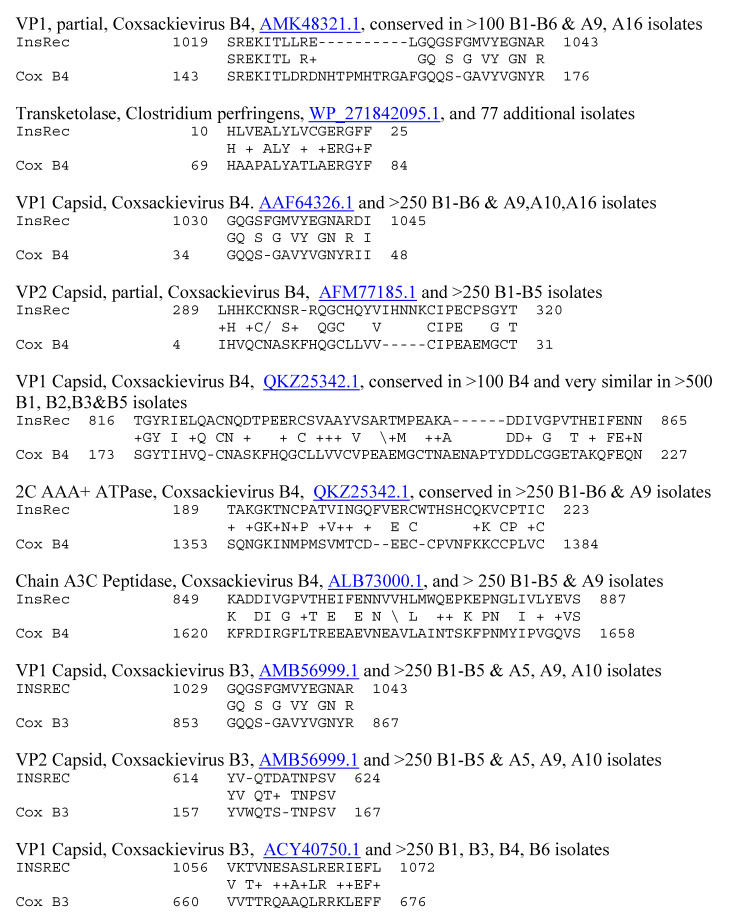
Similarities between coxsackievirus B3 (Cox B3) and the human insulin receptor (INSREC). UniProtKB identifiers are underlined.

**Figure 11 ijms-25-01797-f011:**
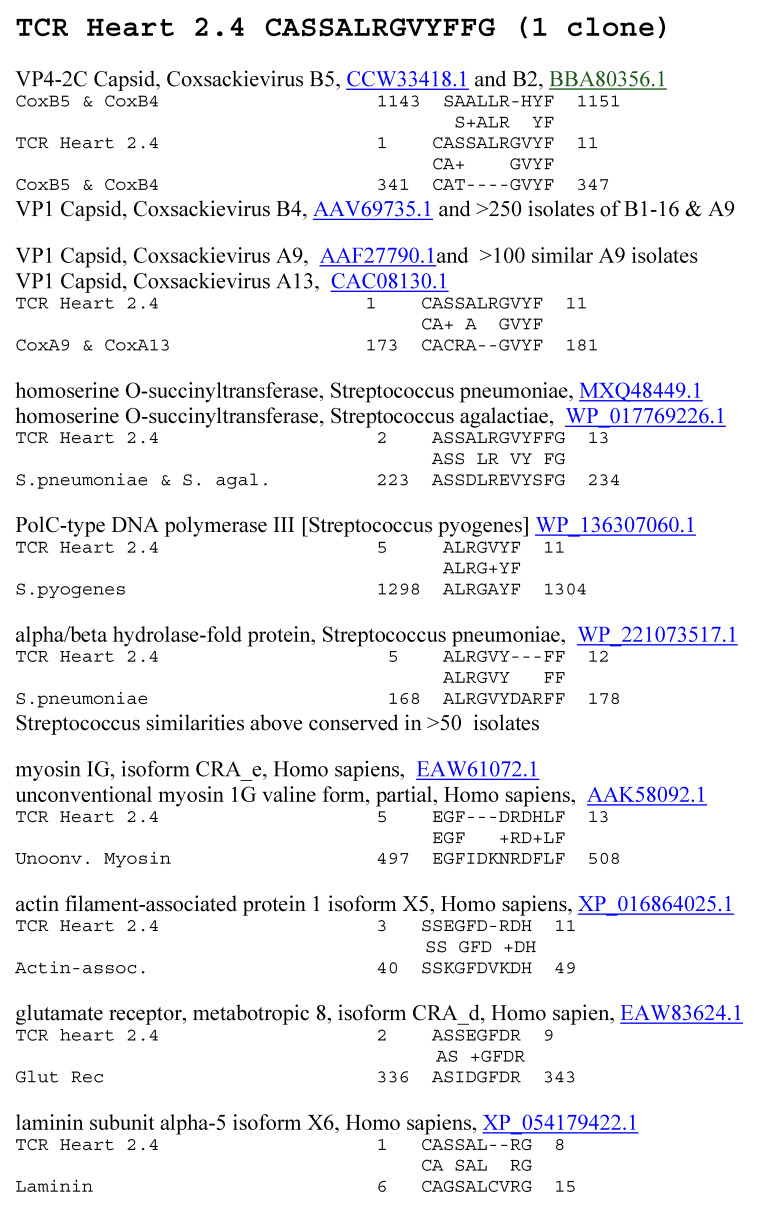
Significant similarities between the TCR Heart 2.4 sequence and the antigens listed in Table 2 above. Underlined numbers are the UniProtKB identifiers.

**Figure 12 ijms-25-01797-f012:**
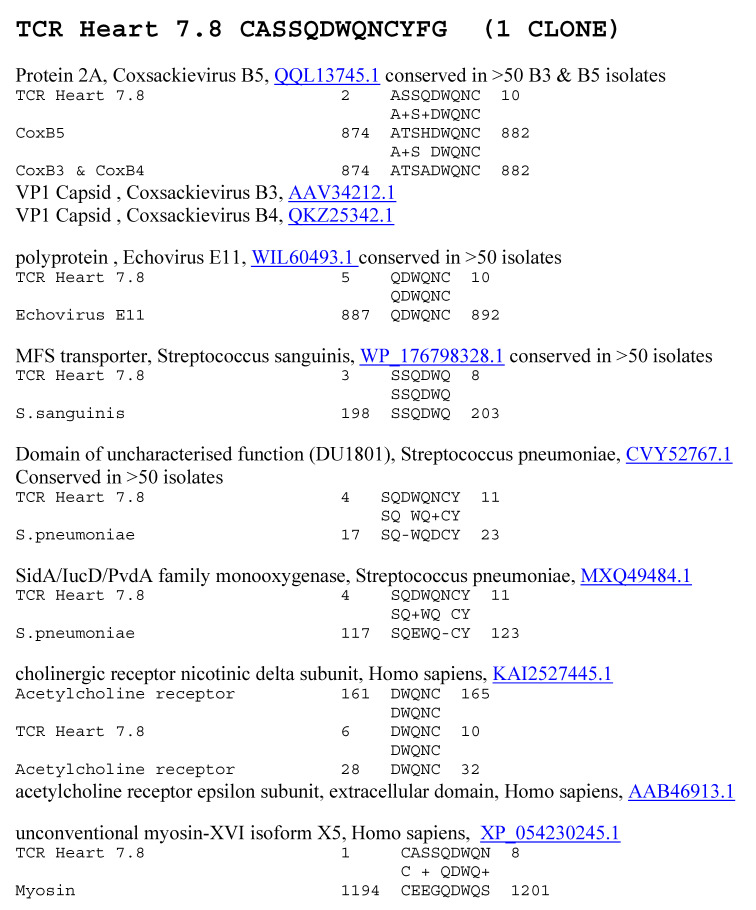
Significant similarities between the TCR Heart 7.8 sequence and the antigens listed in Table 2 above. Underlined numbers are the UniProtKB identifiers.

**Figure 13 ijms-25-01797-f013:**
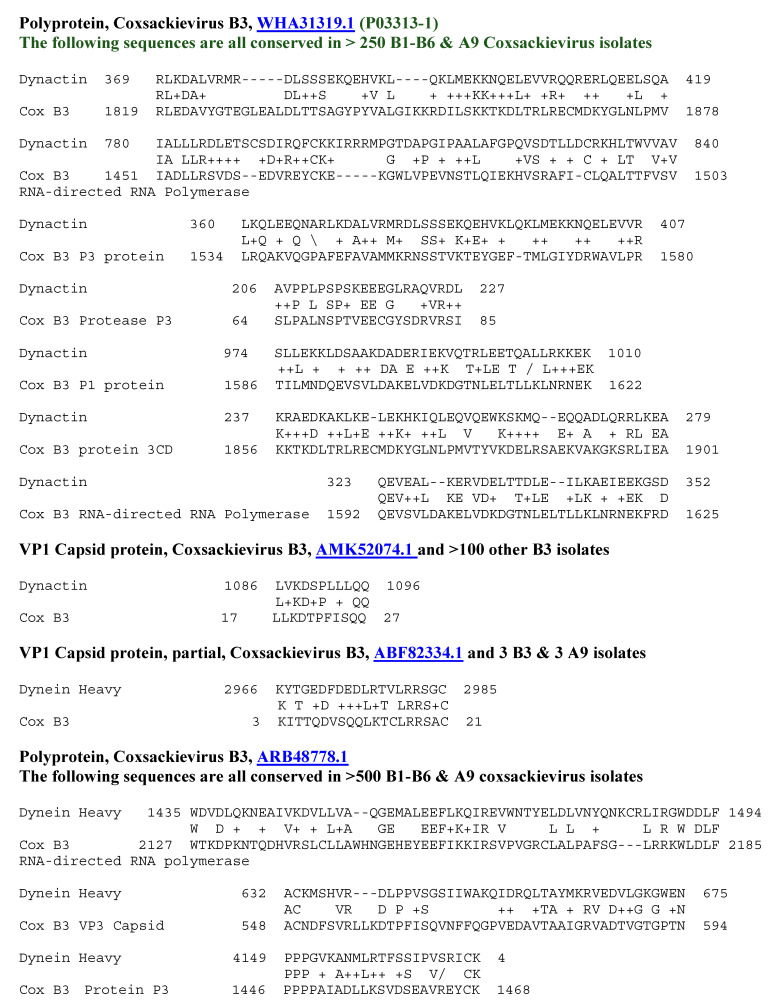
Sequence similarities between coxsackievirus B3 and human dynactin revealed by LALIGN. UniProtKB identifiers are underlined.

**Figure 14 ijms-25-01797-f014:**
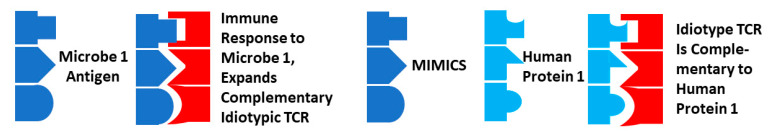
Summary of mimicry and complementarity relationships predicted for T-cell receptors (TCRs) by the molecular mimicry theory.

**Figure 15 ijms-25-01797-f015:**
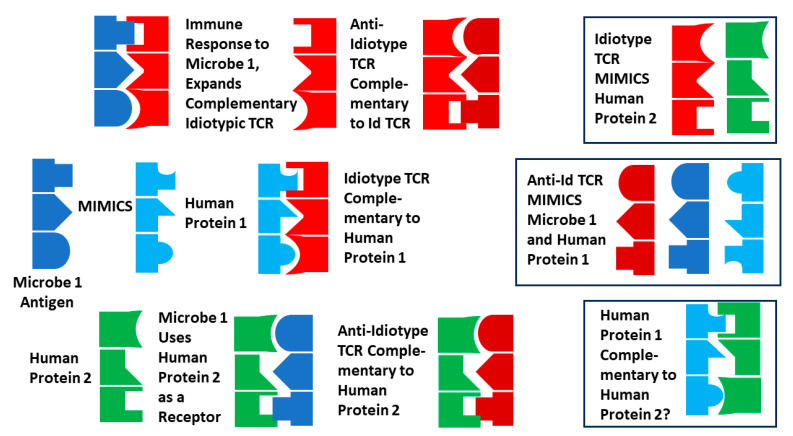
Summary of mimicry and complementarity relationships predicted for T-cell receptors (TCRs) by the anti-idiotype theory.

**Figure 16 ijms-25-01797-f016:**
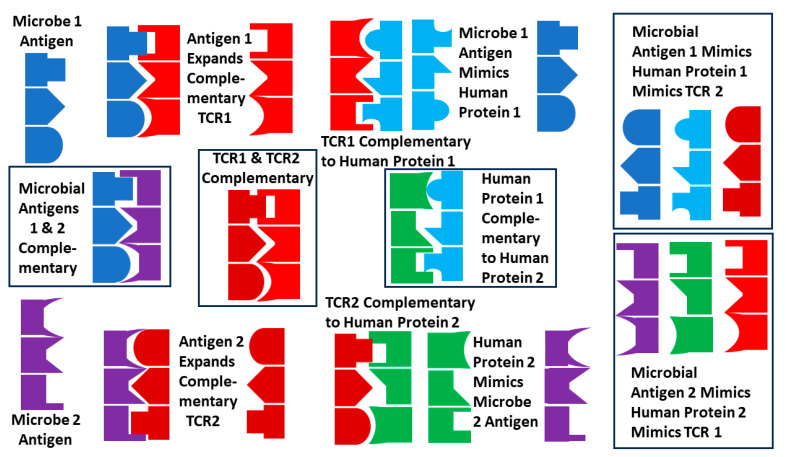
Summary of mimicry and complementarity relationships predicted for T-cell receptors (TCRs) by the complementary antigen theory.

**Figure 17 ijms-25-01797-f017:**
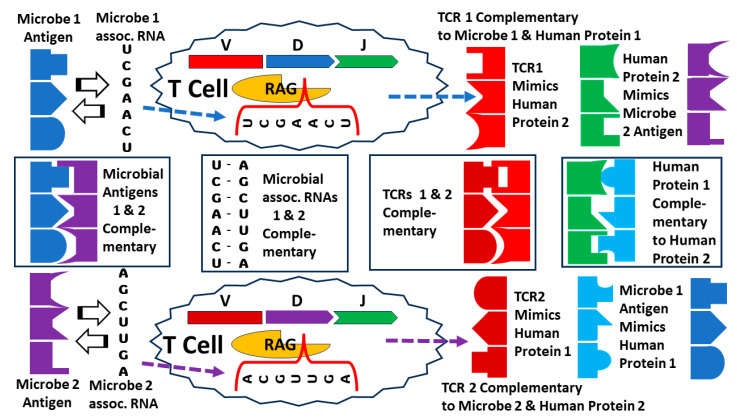
Summary of mimicry and complementarity relationships predicted for T-cell receptors (TCRs) by the antigen template theory in the case of complementary antigens. Note that the antigen template theory does not resolve any of the problems applying the molecular mimicry or anti-idiotype theories to the TCR mimicry data presented in the Results here.

**Table 1 ijms-25-01797-t001:** Summary of significant similarities found using BLASTP between T-cell receptor (TCR) sequence derived from diabetic patients (see Appendix C for sequences A1 through K2.16) to coxsackievirus (Cox), Clostridia (Clost), and various “self”-antigens associated with diabetes and myocarditis. X = significant similarity; blank = no significant similarity (see Figure 6 and Figure 7 and Supplement A for original data). InsRec = insulin receptor; Ins = insulin; GlucRec = glucagon receptor; Gluc = glucagon; PTPRN = receptor-type tyrosine-protein phosphatase-like N, also called “PTP-IA-2”; GAD = glutamic acid decarboxylase; diab = diabetic proteins.

TCR	Cox A	Cox B	Clost	InsRec	Ins	GlucRec	Gluc	PTPRN	GAD	Other Diab	Myosin	Actin
A1	X	X		X		X						
A2		X	X	X		X						
A3		X		X					X		X	
A4		X						X				
A5			X			X					X	
A6	X	X	X	X			X		X			
A7			X			X					X	
A8	X	X	X	X		X						
A9		X	X	X								
A10	X	X	X	X								
A11	X		X					X				
DIA 1		X	X	X			X	X	X			
DIA 2		X	X	X		X					X	
DIA 3	X		X	X		X				X	X	
DIA 4				X		X						
DIA 5		X	X	X		X						
DIA 6	X		X			X			X			
DIA 7	X		X					X		X		
DIA 8	X		X	X	X	X					X	
DIA 9				X	X	X						
DIA 10	X	X	X	X					X			
DIA 11	X		X					X	X		X	
K2.4	X		X	X			X		X		X	
K2.12	X		X	X					X			
K2.16		X	X	X				X				

**Table 2 ijms-25-01797-t002:** Summary of significant similarities found using BLASTP between T-cell receptor (TCR) sequence derived from autoimmune myocarditis patients (see Appendix D for sequences 2.1 through 7.16) to coxsackievirus (Cox), Clostridia (Clost), and various “self”-antigens associated with diabetes and myocarditis. X = significant similarity; (X) = significant similarity to actin-binding proteins rather than actin (ACT) itself; blank = no significant similarity (see Figure 11 and Figure 12 and Supplement B for original data). GAS = group A Streptococci; GBS = group B Streptococci; Strep Virid = Viridians Streptococci; Myo = myosin; Coll = collagen; Lam = laminin; Dyn = dynein; Tit = titan; Ryan Rec = ryanodine receptor; ACh Rec = acetylcholine receptor; Glut = glutamate receptor; Adr = adrenergic receptor; Ins = insulin; Glu = glucagon; Other Cardiac = other cardiac proteins not listed previously.

TCR	Cox A	Cox B	GAS	GBS	Strep Virid	Myo	Act	Coll	Lam	Dyn	Tit	Ryan Rec	ACh Rec	Glut Rec	Adr Rec	Ins	Glu	Other Cardiac
2.1	X		X			X		X	X							X		X
2.2	X		X	X	X			X				X						
2.3	X		X		X													
2.4	X	X	X			X	(X)		X					X				
2.5			X	X	X	X	(X)							X				
2.6	X	X	X			X				X				X				
2.7			X	X		X	(X)											
2.8			X		X	X			X	X								
2.9		X	X			X	(X)											
2.10	X		X		X	(X)		X		X								
2.11	X		X			X	(X)											
2.12	X		X	X		(X)				X								
2.13	X		X			(X)	(X)	X										
2.14	X		X															
2.15		X	X															
2.16		X	X	X			(X)			X								
2.17			X				(X)											
2.18					X	X	(X)			X								
2.19	X	X	X			X		X										
7.1			X			X						X						X
7.2			X	X	X	X		X										X
7.3	X		X	X	X	X												
7.4	X		X	X	X	X	(X)	X		X								
7.5	X	X	X	X	X													
7.6			X	X		X												
7.7	X	X	X		X		(X)											
7.8		X	X		X	X							X					
7.9			X	X														
7.10	X		X			X												
7.11			X	X						X								
7.12			X		X					X								
7.13	X	X	X	X				X										
7.14	X		X															X
7.15	X	X	X	X	X	X	X											
7.16			X	X						X								

**Table 3 ijms-25-01797-t003:** Summary of the binding of diabetic T-cell receptor (TCR) sequences to each other and to diabetes-associated antigens. Const. = constant (Kd); INS = insulin; GLUC = glucagon; InsRec = insulin receptor; Ab = antibody. Data summarized from [130].

Binding Const. (µM)	TCR 1	TCR2, K2.16	TCR4 K2.4	TCR 8	TCR 9	TCR 10	TCR4,8,9	INS	GLUC	InsRec 105–118	InsRec 897–915	INS Ab	GLUC Ab	InsRec α Ab
K2.12	90	130	250	110	110	110	180	75	120	120	>1000	220	110	24
1		110	310	70	70	150	33	80	>10,000	125	300	41	63	28
2, K2.16			230	220	330	120	>10,000	>1000	>10,000	>1000	>1000	55	99	30
4, K2.4				400	470	110	>1000	15	>1000	>1000	>1000	550	>1000	3.8
8					400	270	200	130	140	120	130	>1000	>1000	88
9						>1000	>1000	23	140	150	145	>1000	2.2	>1000
10							90	130	90	110	140	33	>1000	>1000
4,8,9								>1000	>1000	>1000	>1000	12	5.0	300

**Table 4 ijms-25-01797-t004:** Summary of binding constants (Kd) for the association of various group A *Streptococci* (GAS) antibodies to coxsackievirus B3 and B4 (CX) anti-sera and monoclonal antibodies (MABs) and both sets of antibodies and anti-sera to myocarditis-related antigens. Data summarized from [73,95].

Kd	HS CXB4	MN CXB3	CXB3 MAB948	Myosin	Laminin	Actin	Collagen IV	Vitronectin
GAS MA1-10698	4 × 10^−11^	5 × 10^−9^	>10^−5^	2 × 10^−10^	2 × 10^−9^	2 × 10^−6^	1 × 10^−7^	3 × 10^−8^
GAS MA1-10699	6 × 10^−11^	6 × 10^−9^	>10^−5^	>10^−5^	>10^−5^	>10^−5^	>10^−5^	>10^−5^
GAS MA1-10700	8 × 10^−11^	4 × 10^−9^	>10^−5^	3 × 10^−10^	8 × 10^−6^	>10^−5^	3 × 10^−7^	>10^−5^
GAS MA1-10701	3 × 10^−11^	5 × 10^−9^	>10^−5^	>10^−5^	>10^−5^	>10^−5^	>10^−5^	>10^−5^
GAS MBS190189	5 × 10^−10^	1 × 10^−7^	>10^−5^	3 × 10^−8^	4 × 10^−9^	>10^−5^	>10^−5^	>10^−5^
CXB4 Horse				3 × 10^−8^	3 × 10^−7^	2 × 10^−9^	2 × 10^−11^	>10^−5^
CXB3 Monkey				3 × 10^−8^	>10^−5^	7 × 10^−10^	3 × 10^−10^	>10^−5^
CXB3 MAB948				5 × 10^−8^	1 × 10^−7^	>10^−5^	>10^−5^	>10^−5^

## Data Availability

All relevant data related to this study are provided in the body of the text, in the Appendix B, Appendix C and Appendix D, or in the Appendix A.

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
