# Peer review of "T-Cell Receptor Sequences Identify Combined Coxsackievirus–Streptococci Infections as Triggers for Autoimmune Myocarditis and Coxsackievirus–Clostridia Infections for Type 1 Diabetes"

_ijms, 2024, doi:10.3390/ijms25031797_

Round 1

Reviewer 1 Report

Comments and Suggestions for Authors

The hypothesis of this article is that T cell receptor (TCR) sequences expanded in type 1 diabetes (T1D) and autoimmune myocarditis (AM) mimic sequences present in the proteins of the infectious triggers of these diseases.  The infectious trigger examined is coxsackievirus and the author notes that in T1D expanded TCR sequences also mimic Clostridia sequences and, in AM, they mimic Streptococcal antigens.  The author found coxsackievirus sequences that had significant similarity to insulin receptor sequences, Clostridium perfringens sequences that had significant similarity to insulin sequences and expanded TCR sequences from cases of T1D with similarity to both the coxsackievirus, Clostridium and diabetic autoantigens. A similar pairing of bacterial and coxsackievirus peptides and autoantigens from myocarditis is also presented.  The author proposes a model in which each microbial infection expands TCR sequences which can bind self antigens which interact (insulin: insulin receptor) and results in idiotype-anti-idiotype pairs.  This is proposed to lead to a breakdown of immune tolerance resulting in autoimmune effects. 

The author uses TCR sequences from patients with T1D (with similarity to pancreatic autoantigen sequences) and AM (with similarity to cardiac autoantigens) and a control population to identify microbial triggers of T1D and AM. The author identified 25 TCR identified from cases of T1D and 34 from cases of AM.  It is not clear how frequently those with similarity to pancreatic autoantigens were represented in the population of T1D patients used. The 34 TCR sequences from AM were from 1 case of myocarditis and 1 case of dilated cardiomyopathy, which both had detectable coxsackievirus B3 or B4 antigens present in the heart but no viral RNA was detected. Present studies have determined that persistence of coxsackieviruses in dilated cardiomyopathy is linked to the persistence of a very low level of viral replication of a defective form of these viruses in a percentage of these cases (Glenet et al. Sci Rep. 2020 10(1):11947. doi: 10.1038/s41598-020-67648-5). As both of the patients used in the present study had detectable enterovirus in the myocardium,  a low but active replication of virus in the myocardium was present when the lymphocyte were taken for cloning and sequencing the TCR.  It is also likely that these patients also had some degree of autoimmunity to cardiac antigens given the persistent infection but this was not tested for these patients.  I do not think that these TCRs should be labeled as from autoimmune myocarditis but rather as from viral myocarditis.  I note that the references in lines 107-110: “Epidemiological studies demonstrate that 65-95 percent of RHD and AM cases presented with concurrent GAS and CV infections, while uncomplicated CV or GAS infections were almost never associated with development of RHD or AM [74-81]” do not really give that data.  Most of these are an investigation of RHD cases in which the serology indicates that the patient had a prior infection with an enterovirus which means they are not definitely concurrent as most enterovirus infections do not lead to persistent enterovirus infections of the heart (thank goodness!) although they do lead to immunity to the virus.  Most researchers in the field of myocarditis agree that multiple virus infections may exist to a significant degree in patients with myocarditis so they would agree that there are multiple causes of myocarditis in human beings. 

          35% of the T1D TCR sequences have a match to coxsackievirus B (CVB) proteins and 5% to coxsackievirus A (CVA) proteins and 30% of the myocarditis TCR to CVB proteins and 15% to CVA proteins in Figure 1.  This is somewhat simplified as the proteins identified with similarity (in the text and supplementary data) are largely from the enterovirus B species which includes some coxsackievirus A types as well as echoviruses and more recent numbered enterovirus B types. Most of those enterovirus proteins identified with similarity to TCR sequences are to structural proteins which have a higher degree of conservation between strains with the enterovirus B species.  The author should identify the viral sequence by both strain and location in the viral genome and the degree to which this is a conserved sequence (in the type and in the species).  For example, in Figure 7, ADW53828.1 is the sequence of a CVB3 strain adapted in cell culture which differs in 1 position from the majority of human isolates of CVB3 in the capsid protein VP3 sequence with similarity to TCR DIA 4; AKD28015.1 is a CVB5 from Kuwait which has a variation at this site in VP3 from most CVB5 isolates, the second site in the structural protein 2C is fairly conserved in enterovirus B types; 7YMS  is part of the light chain of 9B5, a monoclonal antibody that binds to the neutralizing site

of a strain of coxsackievirus A16 including VP1 97T, 100T, and 103T. The CDRs of this fab are 27-31, 44-55, and 91-96 which do not include these sequences in the figure. I can look at the enterovirus sequences as I have some expertise with this but the lack of significance of the identified sequences of the enterovirus epitopes makes me worry about the significance of the Clostridium sequences.

I appreciate the difficulties of finding TCR sequences from patients with these diseases (as stated by the author) limits the significance of the correlations that the author is drawing.  This is further complicated by the fact that the data available to him (as in GenBank) is not being utilized to demonstrate that he is not defining the viral sequences to sequences that patients are likely to encounter in enterovirus infections (the conserved sites).  I also have doubts that the patients are likely to have concurrent infections with enteroviruses and Streptococci or Clostridium based on the references given. 

Minor points:

In Figure 1, 2 and 3, the use of a line graph to demonstrate the amount of TCR sequences that match virus or bacterial sequences is not appropriate since the viruses or bacteria are not arranged by evolutionary relationship or genetic identity.  Please convert these to bar graphs.

Author Response

Reviewer #1

Open Review

Quality of English Language

( ) I am not qualified to assess the quality of English in this paper
( ) English very difficult to understand/incomprehensible
( ) Extensive editing of English language required
( ) Moderate editing of English language required
( ) Minor editing of English language required
(x) English language fine. No issues detected

Yes

Can be improved

Must be improved

Not applicable

Does the introduction provide sufficient background and include all relevant references?

( )

(x)

( )

( )

Are all the cited references relevant to the research?

( )

( )

(x)

( )

Is the research design appropriate?

( )

( )

(x)

( )

Are the methods adequately described?

( )

(x)

( )

( )

Are the results clearly presented?

( )

(x)

( )

( )

Are the conclusions supported by the results?

( )

( )

(x)

( )

Comments and Suggestions for Authors

The hypothesis of this article is that T cell receptor (TCR) sequences expanded in type 1 diabetes (T1D) and autoimmune myocarditis (AM) mimic sequences present in the proteins of the infectious triggers of these diseases.  The infectious trigger examined is coxsackievirus and the author notes that in T1D expanded TCR sequences also mimic Clostridia sequences and, in AM, they mimic Streptococcal antigens.  The author found coxsackievirus sequences that had significant similarity to insulin receptor sequences, Clostridium perfringens sequences that had significant similarity to insulin sequences and expanded TCR sequences from cases of T1D with similarity to both the coxsackievirus, Clostridium and diabetic autoantigens. A similar pairing of bacterial and coxsackievirus peptides and autoantigens from myocarditis is also presented.  The author proposes a model in which each microbial infection expands TCR sequences which can bind self antigens which interact (insulin: insulin receptor) and results in idiotype-anti-idiotype pairs.  This is proposed to lead to a breakdown of immune tolerance resulting in autoimmune effects. 

Let me begin by thanking the Reviewer for their unusually precise and insightful comments which I have found to be extremely useful in clarifying the limitations of my study and what needs to be tested further.

The author uses TCR sequences from patients with T1D (with similarity to pancreatic autoantigen sequences) and AM (with similarity to cardiac autoantigens) and a control population to identify microbial triggers of T1D and AM. The author identified 25 TCR identified from cases of T1D and 34 from cases of AM.  It is not clear how frequently those with similarity to pancreatic autoantigens were represented in the population of T1D patients used.

I have now added this information into the text at an appropriate spot in the Results.

The 34 TCR sequences from AM were from 1 case of myocarditis and 1 case of dilated cardiomyopathy, which both had detectable coxsackievirus B3 or B4 antigens present in the heart but no viral RNA was detected. Present studies have determined that persistence of coxsackieviruses in dilated cardiomyopathy is linked to the persistence of a very low level of viral replication of a defective form of these viruses in a percentage of these cases (Glenet et al. Sci Rep. 2020 10(1):11947. doi: 10.1038/s41598-020-67648-5).

The Reviewer has either mis-read or mis-remembered the nature of the cited study, the complete title of which, notably, is: “Major 5'terminally deleted enterovirus populations modulate type I IFN response in acute myocarditis patients and in human cultured cardiomyocytes.” There is no mention of chronic dilated cardiomyopathy patients in the study group. Whether such defective virus production is present in autoimmune myocarditis is not addressed nor can I find any studies suggesting that it is.

As both of the patients used in the present study had detectable enterovirus in the myocardium, a low but active replication of virus in the myocardium was present when the lymphocyte were taken for cloning and sequencing the TCR. 

This statement is also incorrect. There was NO evidence of active replication in the myocardium: The paper states clearly that, “RT-PCR analysis for detection of viral genome did not show evidence of EV RNA in any of the heart specimens studied.” (Results, paragraph 2, page 197).  The positive results were from “mAbs specific for CVB1, 3, 4 and 5 serotypes” but in light of the cross-reactivity of such antibodies for cardiac “self” proteins due to molecular mimicry demonstrated in previous studies that we have carried out, these results are, at best, ambiguous and may just show cross-reactivity between CVBs and myocardial proteins.

It is also likely that these patients also had some degree of autoimmunity to cardiac antigens given the persistent infection but this was not tested for these patients.

Exactly the point just made. So, is this an objection or a statement of agreement with the direction of my manuscript? The  Reviewer seems to argue above that these are not autoimmune myocarditis cases but now says that autoimmunity is probably present – one can’t have it both ways. Secondly, the Reviewer focuses solely on the presence of coxsackievirus but ignores the multiple lines of evidence (antibody studies and the present TCR studies) that demonstrate the presence of immune stimulation against Streptococcal antigens in AM patients. This paper fully agrees that coxsackieviruses (or enteroviruses more generally) are essential to AM etiology but address two questions that the Reviewer does not: 1) why do coxsackievirus infections result in myocarditis in some patients, T1DM in others, and no autoimmunity in the vast majority? And 2) why are Strep-like TCR expanded in AM while Clostridia-like TCR are expanded in T1DM? At best, the Reviewer’s objections simply add to these conundrums without pointing toward any resolution of them.

 I do not think that these TCRs should be labeled as from autoimmune myocarditis but rather as from viral myocarditis. 

Obviously, we disagree. The key point is that these patients display expansion of both coxsackievirus and Strep-mimicking TCR. If coxsackievirus infection was the only cause of these patients’ symptoms, how does one explain the evidence pointing toward Strep? The Reviewer does not address why this should be so.

I note that the references in lines 107-110: “Epidemiological studies demonstrate that 65-95 percent of RHD and AM cases presented with concurrent GAS and CV infections, while uncomplicated CV or GAS infections were almost never associated with development of RHD or AM [74-81]” do not really give that data.  Most of these are an investigation of RHD cases in which the serology indicates that the patient had a prior infection with an enterovirus which means they are not definitely concurrent…

I agree that I over-interpreted the data in references 74-81 and have modified the language where these references are cited to indicate, as the Reviewer says, that these studies only indicate that these patients recently had infections with enteroviruses resulting in increased antibody titers, which could be evidence of prior, overlapping or concomitant infections. I have also included this problem as one that needs to be addressed in experimental and clinical tests of the hypothesis presented in this paper and also as a limitation of the current study.

…as most enterovirus infections do not lead to persistent enterovirus infections of the heart (thank goodness!) although they do lead to immunity to the virus.  Most researchers in the field of myocarditis agree that multiple virus infections may exist to a significant degree in patients with myocarditis so they would agree that there are multiple causes of myocarditis in human beings. 

            35% of the T1D TCR sequences have a match to coxsackievirus B (CVB) proteins and 5% to coxsackievirus A (CVA) proteins and 30% of the myocarditis TCR to CVB proteins and 15% to CVA proteins in Figure 1.  This is somewhat simplified as the proteins identified with similarity (in the text and supplementary data) are largely from the enterovirus B species which includes some coxsackievirus A types as well as echoviruses and more recent numbered enterovirus B types. Most of those enterovirus proteins identified with similarity to TCR sequences are to structural proteins which have a higher degree of conservation between strains with the enterovirus B species.  The author should identify the viral sequence by both strain and location in the viral genome and the degree to which this is a conserved sequence (in the type and in the species).  For example, in Figure 7, ADW53828.1 is the sequence of a CVB3 strain adapted in cell culture which differs in 1 position from the majority of human isolates of CVB3 in the capsid protein VP3 sequence with similarity to TCR DIA 4; AKD28015.1 is a CVB5 from Kuwait which has a variation at this site in VP3 from most CVB5 isolates, the second site in the structural protein 2C is fairly conserved in enterovirus B types;

Done!

 I should note that I have swapped out ADW53828.1 with one of the hundreds of naturally occurring strains, which does result in the loss of one amino acid identity but still leaves the conserved similarities within the range of significance established for the study. Indeed, some strains contain variations that are highly conserved, while other contains ones that are unique or found in only a few strains. I have now discussed this fact in the text, pointing out that, as the Reviewer seems to be arguing, some of the similarities identified by the TCR similarity method used here probably represent more significant autoimmune disease risks than others. On the other hand, one cannot discount the possibility that strains displaying rare similarities may be the most dangerous in terms of autoimmune disease risk.

 7YMS  is part of the light chain of 9B5, a monoclonal antibody that binds to the neutralizing siteof a strain of coxsackievirus A16 including VP1 97T, 100T, and 103T. The CDRs of this fab are 27-31, 44-55, and 91-96 which do not include these sequences in the figure.

I have deleted 7YMS, which, since it is an antibody as the Reviewer notes; it should not have been listed.

I can look at the enterovirus sequences as I have some expertise with this but the lack of significance of the identified sequences of the enterovirus epitopes makes me worry about the significance of the Clostridium sequences.

I have performed the same analysis of how frequently each Clostridium similarity is conserved as I have performed for the enteroviruses and added that data to the appropriate figures.

I appreciate the difficulties of finding TCR sequences from patients with these diseases (as stated by the author) limits the significance of the correlations that the author is drawing.  This is further complicated by the fact that the data available to him (as in GenBank) is not being utilized to demonstrate that he is not defining the viral sequences to sequences that patients are likely to encounter in enterovirus infections (the conserved sites).  I also have doubts that the patients are likely to have concurrent infections with enteroviruses and Streptococci or Clostridium based on the references given. 

The underlined sentence above is so confusing with its multiple negatives that I’m not sure whether I have addressed the point adequately above. I hope so! If not, please explain further and I’ll try again.

I understand the doubts about the concurrent infections and have made the limitations of the cited studies clearer regarding this point. I do think that the fact that enterovirus antibody titers are higher in RHD patients does suggest at least recent infection (making overlap of the immune responses to both Strep and enteroviruses much more likely) and one still has to address why Strep and enterovirus TCR are both very significantly expanded when TCR mimicking no other virus or bacterium is similarly expanded in these patients.

However, in the end, what I have presented is a testable hypothesis and the acid test is whether future clinical studies explicitly looking for evidence of overlapping or concurrent infections actually find them. This may not be the explanation for the antibody and TCR data, but then some other hypothesis will be needed, which will, in itself, be a step forward.

Minor points:

In Figure 1, 2 and 3, the use of a line graph to demonstrate the amount of TCR sequences that match virus or bacterial sequences is not appropriate since the viruses or bacteria are not arranged by evolutionary relationship or genetic identity.  Please convert these to bar graphs.

Done!

Submission Date

09 November 2023

Date of this review

07 Dec 2023 00:01:30

Reviewer 2 Report

Comments and Suggestions for Authors

The article written by Root-Bernstein entitled "T Cell Receptor Sequences Identify Combined Coxsackievirus-Streptococci Infections as Triggers for Autoimmune Myocarditis and Coxsackievirus-Clostridia Infections for Type 1 Diabetes" is well presented and easy to understand. It is from the rare publications discussing the role of mimicry theory events that are responsible of the pathogenesis of autoimmune diseases, i.e, T1D and cardiomyopathy in this case.

The author use many methods and elaborate a well-characterized theories to explain the role of pathogen protein mimiting the TCR in the autoimmune diseases.

However, some remarks that can improve the presentation and the quality of the article:

* References used: Author cited a lot his references. It is normal because they are all in the main objective of the study. But he forget others references that are also relevant to the study and I suggest him to add some recent references especially in the introduction:

- Cloning and Molecular Characterization of the Recombinant CVB4E2 Immunogenic Viral Protein (rVP1), as a Potential Subunit Protein for Vaccine and Immunodiagnostic Reagent Candidate. Microorganisms2023, 11(5), 1192

Viral Protein VP1 Virus-like Particles (VLP) of CVB4 Induces Protective Immunity against Lethal Challenges with Diabetogenic E2 and Wild Type JBV Strains in Mice Model. Viruses, 2023, 15(4), 878

The introduction of mutations in the wild type coxsackievirus B3 (CVB3) IRES RNA leads to different levels of in vitro reduced replicative and translation efficiencies. PLoS ONE, 2022, 17(10 October), e0274162

* Presentation of Results: It is strong recommended to put all Figures 6, 7, 8, 9, 10, 11, 12 and 13 in the Appendix section. That will make the results contents more easy to read and heavy to fellow by readers. Others figures in Results are very relevant and well presented and they are enough to understand the advanced theory.

This link is disabled., 2022, 17(10 October), e0274162

This link is disabled 2023, 15(4), 878

Author Response

Reviewer #2

Open Review

Quality of English Language

(x) I am not qualified to assess the quality of English in this paper
( ) English very difficult to understand/incomprehensible
( ) Extensive editing of English language required
( ) Moderate editing of English language required
( ) Minor editing of English language required
( ) English language fine. No issues detected

Yes

Can be improved

Must be improved

Not applicable

Does the introduction provide sufficient background and include all relevant references?

(x)

( )

( )

( )

Are all the cited references relevant to the research?

( )

(x)

( )

( )

Is the research design appropriate?

(x)

( )

( )

( )

Are the methods adequately described?

(x)

( )

( )

( )

Are the results clearly presented?

( )

(x)

( )

( )

Are the conclusions supported by the results?

(x)

( )

( )

( )

Comments and Suggestions for Authors

The article written by Root-Bernstein entitled "T Cell Receptor Sequences Identify Combined Coxsackievirus-Streptococci Infections as Triggers for Autoimmune Myocarditis and Coxsackievirus-Clostridia Infections for Type 1 Diabetes" is well presented and easy to understand. It is from the rare publications discussing the role of mimicry theory events that are responsible of the pathogenesis of autoimmune diseases, i.e, T1D and cardiomyopathy in this case.

The author use many methods and elaborate a well-characterized theories to explain the role of pathogen protein mimiting the TCR in the autoimmune diseases.

However, some remarks that can improve the presentation and the quality of the article:

* References used: Author cited a lot his references. It is normal because they are all in the main objective of the study. But he forget others references that are also relevant to the study and I suggest him to add some recent references especially in the introduction:

- Cloning and Molecular Characterization of the Recombinant CVB4E2 Immunogenic Viral Protein (rVP1), as a Potential Subunit Protein for Vaccine and Immunodiagnostic Reagent Candidate. Microorganisms, 2023, 11(5), 1192

- Viral Protein VP1 Virus-like Particles (VLP) of CVB4 Induces Protective Immunity against Lethal Challenges with Diabetogenic E2 and Wild Type JBV Strains in Mice Model. Viruses, 2023, 15(4), 878

- The introduction of mutations in the wild type coxsackievirus B3 (CVB3) IRES RNA leads to different levels of in vitro reduced replicative and translation efficiencies. PLoS ONE, 2022, 17(10 October), e0274162

This link is disabled., 2022, 17(10 October), e0274162

This link is disabled 2023, 15(4), 878

I have added these reference, but not in the Introduction since they are not appropriate there. Instead I have added a new, brief section on the importance of the present work for vaccine development, which is a more appropriate place to cite these references.

* Presentation of Results: It is strong recommended to put all Figures 6, 7, 8, 9, 10, 11, 12 and 13 in the Appendix section. That will make the results contents more easy to read and heavy to fellow by readers. Others figures in Results are very relevant and well presented and they are enough to understand the advanced theory.

I am unwilling to move these data to Appendices. As the comments from the first Reviewer make clear, the “devil is in the details” when it comes to evaluating TCR-microbial similarities and some readers will want to see these details. In addition, moving all of the Figures requested by this Reviewer would essentially leave nothing but summary information, which I personally would find unacceptable. The only way to evaluate a paper is to examine the actual data! I’m sorry if this makes the paper less readable than it could be, but I don’t expect my readers simply to accept my summaries and conclusions.

Submission Date

09 November 2023

Date of this review

26 Dec 2023 10:26:17

Reviewer 3 Report

Comments and Suggestions for Authors

The author explored the connection betweenTCR sequences and triggers for autoimmune myocarditis and type 1 diabetes by a special method. By bioinformatical methods, the author revealed that TCR sequences from patients with autoimmune myocarditis and type 1 diabetes mimicked viral and bacterial antigens. He  suggested that combined coxsackievirus-streptococci infections triggered autoimmune myocarditis, while coxsackievirus-clostridia infections trigger type 1 diabetes, by the mechanism of TCR mimic. The author suggested that TCR sequences expanded during infections can be used to identify the triggering agents involved in autoimmune diseases. This is a very valuable study because understanding the role of TCR sequences in autoimmune diseases can lead to the development of new treatments and prevention strategies. Addtionally, the methods used in this study is very interesting.

Here are some suggestions and questions.

1. In the part of introduction (eg. Line 37), it is better for the author to provide a more detailed explanation of what is meant by "mimic these viral antigens." The author would be better off moving some parts from the discussion to the introduction, as this would allow for an early explanation of some fundamental concepts of the research, making it more reader-friendly.

2. The data provided by the author clearly demonstrated that some TCRs do indeed mimic some antigens associated with autoimmune diseases, but the author needed to discuss the next step possibility of using experimental methods to test whether there are actually immune responses targeting these TCRs.

Author Response

Reviewer #3

Top of Form

Open Review

Quality of English Language

( ) I am not qualified to assess the quality of English in this paper
( ) English very difficult to understand/incomprehensible
( ) Extensive editing of English language required
( ) Moderate editing of English language required
( ) Minor editing of English language required
(x) English language fine. No issues detected

Yes

Can be improved

Must be improved

Not applicable

Does the introduction provide sufficient background and include all relevant references?

(x)

( )

( )

( )

Are all the cited references relevant to the research?

(x)

( )

( )

( )

Is the research design appropriate?

(x)

( )

( )

( )

Are the methods adequately described?

(x)

( )

( )

( )

Are the results clearly presented?

(x)

( )

( )

( )

Are the conclusions supported by the results?

(x)

( )

( )

( )

Comments and Suggestions for Authors

The author explored the connection between TCR sequences and triggers for autoimmune myocarditis and type 1 diabetes by a special method. By bioinformatical methods, the author revealed that TCR sequences from patients with autoimmune myocarditis and type 1 diabetes mimicked viral and bacterial antigens. He  suggested that combined coxsackievirus-streptococci infections triggered autoimmune myocarditis, while coxsackievirus-clostridia infections trigger type 1 diabetes, by the mechanism of TCR mimic. The author suggested that TCR sequences expanded during infections can be used to identify the triggering agents involved in autoimmune diseases. This is a very valuable study because understanding the role of TCR sequences in autoimmune diseases can lead to the development of new treatments and prevention strategies. Addtionally, the methods used in this study is very interesting.

Here are some suggestions and questions.

Thanks for the useful suggestions!

  1. In the part of introduction (eg. Line 37), it is better for the author to provide a more detailed explanation of what is meant by "mimic these viral antigens." The author would be better off moving some parts from the discussion to the introduction, as this would allow for an early explanation of some fundamental concepts of the research, making it more reader-friendly.

ADDED!

  1. The data provided by the author clearly demonstrated that some TCRs do indeed mimic some antigens associated with autoimmune diseases, but the author needed to discuss the next step possibility of using experimental methods to test whether there are actually immune responses targeting these TCRs.

New, extensive section added in conclusion doing just that!

Submission Date

09 November 2023

Date of this review

28 Dec 2023 14:35:20

Bottom of Form

© 1996-2023 MDPI (Basel, Switzerland) unless otherwise stated

Disclaim

Round 2

Reviewer 1 Report

Comments and Suggestions for Authors

Thank you for your comments on my review of your first version of this paper.

In my previous review, I stated:

“The author uses TCR sequences from patients with T1D (with similarity to pancreatic autoantigen sequences) and AM (with similarity to cardiac autoantigens) and a control population to identify microbial triggers of T1D and AM. The author identified 25 TCR identified from cases of T1D and 34 from cases of AM.  It is not clear how frequently those with similarity to pancreatic autoantigens were represented in the population of T1D patients used.”

The author replied: I have now added this information into the text at an appropriate spot in the results.

I assume this in lines 268-285 which now refers to the number of TCR from the group of patients demonstrated similarity to an autoantigen.  I wished to know how frequently these TCR were present in the group of patients with type 1 diabetes or in the group of patients with myocarditis, in other words, what percentage of patients with the disease had these TCR.  I know the author has used a control set of TCRs from people without these diseases and found statistically significant increases in TCRs from diseases as compared to the TCRS from the control population with similarity to specific microbial antigens.  By reading the publications from which the TCRs were taken, I found that there were only 2 myocarditis patients with the 34 TCRs used in this study and there were 22 patients providing the 25 TCRs used for the type 1 diabetes analysis.  We don’t know how many donors are represented in the 325 TCRs of the control group used by the author.  However, in Seay HR, et al. (JCI Insight. 2016 Dec 8;1(20):e88242. doi: 10.1172/jci.insight.88242; the publication providing the TCRs of 18 type 1 diabetes patients in your study), the authors compare TCRs from these type 1 diabetic donors to TCRs from a control population without type 1 diabetes and found that most of the TCRs from this study used by you to represent TCRs from type 1 diabetics are present in a similar number of the control donors also.  I still do not feel that you have used a sufficient population TCR donors to represent what is commonly present in type 1 diabetics and in myocarditis over the control population.  I do understand the difficulties of generating a sufficient pool of TCR donors but without that data the significance of the results is questionable.

My review of the first version of this paper: “Present studies have determined that persistence of coxsackieviruses in dilated cardiomyopathy is linked to the persistence of a very low level of viral replication of a defective form of these viruses in a percentage of these cases (Glenet et al. Sci Rep. 2020 10(1):11947. doi: 10.1038/s41598-020-67648-5). As both of the patients used in the present study had detectable enterovirus in the myocardium, a low but active replication of virus in the myocardium was present when the lymphocyte were taken for cloning and sequencing the TCR. “

The author replied that the cited study was of acute myocarditis patients and not autoimmune myocarditis.  I suggest looking at the papers in this recent review: Tschöpe C, et al. Myocarditis and inflammatory cardiomyopathy: current evidence and future directions. Nat Rev Cardiol. 2021 Mar;18(3):169-193. doi: 10.1038/s41569-020-00435-x.  Not all myocarditis cases are caused by viruses but viral myocarditis has been shown to have autoimmunity.  Despite the statement in lines 93-95 and repeated on lines 667-669: “CVB3 only induces AM in mice when coinoculated with cardiac myosin or with heart antigens after being passaged through heart tissue (65-67).”  the literature on coxsackievirus B3 in mouse models of myocarditis going back to the 1960s has shown the presence of autoimmunity after viral infection. For example: Fairweather D, Rose NR. Coxsackievirus-induced myocarditis in mice: a model of autoimmune disease for studying immunotoxicity. Methods. (2007) 41:118–22. doi: 10.1016/j.ymeth.2006.07.009; Latva-Hirvela J, et al. Development of troponin autoantibodies in experimental coxsackievirus B3 myocarditis. Eur J Clin Invest. (2009) 39:457–62. doi: 10.1111/j.1365-2362.2009.02113.x; Takata S, et al. Identification of autoantibodies with the corresponding antigen for repetitive coxsackievirus infection-induced cardiomyopathy. Circ J. (2004) 68:677–82. doi: 10.1253/circj.68.677. These models did not require inoculation with cardiac antigens. 

What I was hoping to help you understand is that the two myocarditis cases from 1994 that you used for TCRs in the AM group most likely did have viral myocarditis because they were able to use monoclonal anti-coxsackievirus antibodies to detect coxsackievirus B antigens in the heart tissue.  They probably were not using an RT-PCR capable of detecting the very low level of defective persisting coxsackievirus.  I don’t agree that the detection of the virus in the heart tissue is background staining due to cross reactivity. The monoclonal antibodies used to detect the CVB antigens gave a strong signal in the myocarditis tissues and not in the normal tissues because these monoclonals were selected as antibodies which would not bind uninfected cardiac tissues.  The figures demonstrate this in Luppi et al. 2003 Human Immunology 64: 194-210.  I agree that many cases of myocarditis may develop autoimmunity and I agree that you are using two patients who have enteroviral myocarditis.  Very likely these patients have TCRs related to that infection which you have demonstrated to have reactivity to streptococcal antigens but again this is a very small number of patients. I’m not satisfied that you have sufficient support for your hypothesis using 2 patients who had a coxsackievirus infection of the heart and may or may not have had prior streptococcal infection. 

I will restate the paragraph of my review that the author found difficult to interpret:

I appreciate that the difficulties of finding TCR sequences from patients with myocarditis or with type 1 diabetes limits the significance of the correlations that the author is drawing.  This is further complicated by the fact that the author is not using the data available to him in GenBank to demonstrate that he is defining the cross-reaction to antigens likely to be conserved in enteroviral infections.  If these viral epitopes were in the capsid of the virus, different enterovirus types would generate different TCRs and there would likely be a limited number of types which would be associated with myocarditis (if this mechanism played a large part in generating pathology). I see that the author has noted the viral protein in the second version of this manuscript but has not defined whether these are epitopes conserved or not conserved among the coxsackievirus Bs or enterovirus Bs.

The author has stated in the reply to my previous review: “I do think the fact that enterovirus antibody titers are higher in RHD patients does suggest at least recent infection…).  I do not see this in the literature. It is known that some murine monoclonal antibodies cross react with streptococcal antigens, cardiac valve proteins and coxsackievirus B capsids (Cunningham M, et al. 1992 PNAS 89 (4) 1320-1324). These monoclonal antibodies also are pathogenic to cardiac cells. Are any of the TCRs in the myocarditis patients mimicking the epitopes suggested in this 1992 study? I don’t see any evidence in references 59-62 that enteroviral antibodies are increased in rheumatic fever patients. In the case described in Makaryus et al. (J Infect Dis 2006 14(4): e1-4) post streptococcal vaccination followed by viral (not necessarily enteroviral) upper respiratory infections caused recurrent myocarditis but no increase in viral or streptococcal antibodies was noted.

All in all, the hypothesis of this study is interesting but very limited.  If the author wishes to use this as a basis for proposing further studies, it help if the author made the significance of the figures (number of patients from which the TCRs were obtained, lack of a matched control group) and made the background of these diseases in the literature clear.  It is detracting from the value of this study to make erroneous claims about what is known of myocarditis and type 1 diabetes. 

Comments on the Quality of English Language

There a quite a few typographic errors in the reference numbering.

Author Response

Reviewer 1 #2

Open Review

Quality of English Language

( ) I am not qualified to assess the quality of English in this paper
( ) English very difficult to understand/incomprehensible
( ) Extensive editing of English language required
( ) Moderate editing of English language required
(x) Minor editing of English language required
( ) English language fine. No issues detected

Yes

Can be improved

Must be improved

Not applicable

Does the introduction provide sufficient background and include all relevant references?

( )

( )

(x)

( )

Are all the cited references relevant to the research?

( )

(x)

( )

( )

Is the research design appropriate?

( )

( )

(x)

( )

Are the methods adequately described?

( )

(x)

( )

( )

Are the results clearly presented?

( )

(x)

( )

( )

Are the conclusions supported by the results?

( )

( )

(x)

( )

Comments and Suggestions for Authors

Thank you for your comments on my review of your first version of this paper.

In my previous review, I stated:

“The author uses TCR sequences from patients with T1D (with similarity to pancreatic autoantigen sequences) and AM (with similarity to cardiac autoantigens) and a control population to identify microbial triggers of T1D and AM. The author identified 25 TCR identified from cases of T1D and 34 from cases of AM.  It is not clear how frequently those with similarity to pancreatic autoantigens were represented in the population of T1D patients used.”

The author replied: I have now added this information into the text at an appropriate spot in the results.

I assume this in lines 268-285 which now refers to the number of TCR from the group of patients demonstrated similarity to an autoantigen.  I wished to know how frequently these TCR were present in the group of patients with type 1 diabetes or in the group of patients with myocarditis, in other words, what percentage of patients with the disease had these TCR.  I know the author has used a control set of TCRs from people without these diseases and found statistically significant increases in TCRs from diseases as compared to the TCRS from the control population with similarity to specific microbial antigens.  By reading the publications from which the TCRs were taken, I found that there were only 2 myocarditis patients with the 34 TCRs used in this study and there were 22 patients providing the 25 TCRs used for the type 1 diabetes analysis.

I don’t know how you came up with 22 patients for the T1DM data. Reference 230 lists 3 TCR from one patient; Ref 231 has 7 TCR from 3 patients; and Ref 232 derives data from 18 patients, but I only used the sequences in which the T1DM sequences were statistically dominant, which derived from only 7 of these patients. So, that makes 25 TCR from 13 patients.

 We don’t know how many donors are represented in the 325 TCRs of the control group used by the author.

ADDED: 325 TCR sequences from 135 individuals. I have also added the specific references rather than citing a previous paper of mine so that this figure can be more easily verified and the sequences accessed.

 However, in Seay HR, et al. (JCI Insight. 2016 Dec 8;1(20):e88242. doi: 10.1172/jci.insight.88242; the publication providing the TCRs of 18 type 1 diabetes patients in your study), the authors compare TCRs from these type 1 diabetic donors to TCRs from a control population without type 1 diabetes and found that most of the TCRs from this study used by you to represent TCRs from type 1 diabetics are present in a similar number of the control donors also.

The Reviewer fails to note that these “control” TCR came from type 2, insulin antibody positive and other/Flatbush diabetic patients, most, if not all, of whom were using insulin.  In other words, in the Seay study, many of the TCR sequences would be EXPECTED to occur in both the T1D and “non-T1D” patient groups because all were insulin antibody positive and/or diabetics of some sort. In other words, the “controls” in this study were not healthy individuals such as those I used for my controls but a group made up of other types of diabetes.

 I still do not feel that you have used a sufficient population TCR donors to represent what is commonly present in type 1 diabetics and in myocarditis over the control population.  I do understand the difficulties of generating a sufficient pool of TCR donors but without that data the significance of the results is questionable.

I agree. I would love to have found more TCR to analyze but, as you admit, these do not appear to exist at present.  I list this problem as a limitation several times in the paper, emphasizing it in the “Limitations” section. The paper presents an hypothesis that is clearly testable in many ways, including sequencing more TCRs from more patients (and controls). I’m hoping this study will be an encouragement for such TCR sequencing to get done.

I must also note that despite this Reviewer’s opinion, the manuscript has met the standard for presenting a viable hypothesis according to Reviewers 2 and 3. Obviously there is disagreement on this point but it is not just my opinion versus this Reviewer’s.

My review of the first version of this paper: “Present studies have determined that persistence of coxsackieviruses in dilated cardiomyopathy is linked to the persistence of a very low level of viral replication of a defective form of these viruses in a percentage of these cases (Glenet et al. Sci Rep. 2020 10(1):11947. doi: 10.1038/s41598-020-67648-5). As both of the patients used in the present study had detectable enterovirus in the myocardium, a low but active replication of virus in the myocardium was present when the lymphocyte were taken for cloning and sequencing the TCR. “

The author replied that the cited study was of acute myocarditis patients and not autoimmune myocarditis.  I suggest looking at the papers in this recent review: Tschöpe C, et al. Myocarditis and inflammatory cardiomyopathy: current evidence and future directions. Nat Rev Cardiol. 2021 Mar;18(3):169-193. doi: 10.1038/s41569-020-00435-x. 

To which I reply with a more recent review by Badrinath A, Bhatta S, Kloc A. Persistent viral infections and their role in heart disease. Front Microbiol. 2022 Nov 24;13:1030440. doi: 10.3389/fmicb.2022.1030440. Badrinath, et al. point out that while about thirty percent of myocarditis patients display evidence of persistent, defective virus particles, only about three percent of patients have virus particles that are replication competent. There is no known mechanism by which replication incompetent (otherwise known as “latent”) viruses can stimulate ongoing immunity let alone autoimmunity (e.g., cytomegalovirus and Epstein-Barr virus, which persist in latent forms for decades without causing any adverse affects). This means that about seventy percent of myocarditis patients do not have any residual virus present and ninety-eight percent have no replication viable virus present. Purely on statistical grounds, this makes it very unlikely that the two patients from whom I derived the TCR sequences had active virus infections and it is therefore illogical for this Reviewer to therefore argue that they represent “infectious myocarditis” patients.  Since the authors of that study tested for the presence of virus genome using RT-PCR and failed to find any in either patient, the evidence favors the conclusion that they did not have infectious myocarditis.

I can take this argument further. Let’s assume that the Reviewer is correct that the RT-PCR missed the presence of replication deficient forms of the coxsackievirus. To begin with, this would still fail to make these two patients “infectious myocarditis” patients because their  deficient viruses would have been unable to infect other myocardial cells or cause tissue damage. Moreover, the role of defective viruses in myocarditis is, at this point in time, conjectural. As the Tschöpe, et al. and Badrinath, et al. studies both state explicitly (Badrinath quoted at length below), there is no known mechanism by which such viruses could drive autoimmunity, just speculations. The use of such conjectures by the Reviewer is neither appropriate nor convincing.

Badrinath, et al (2022) relevant passages initalics:

The persistence of enterovirus in the heart tissue and its potential role in cardiomyopathy has been an active area of investigation for decades. The analysis of human EMB samples showed that 25 out of 70 (37.5%) of patients with idiopathic DCM and 21 out of 70 (32.8%) of patients with chronic coronary disease had detectable levels of enterovirus, whereas no EV genome was picked up in the hearts of 45 healthy donors (Andréoletti et al., 2000). Only 2 out of 70 individuals had an RNA replication intermediate, known as antigenomic RNA, present in their samples along with VP1 antigen expression, consistent with active viral replication. The majority of patients (97%) had a latent EV infection, demonstrated by the absence of antigenomic RNA at the time of heart tissue collection (Andréoletti et al., 2000). Evaluation of 172 patients with biopsy-proven viral infection in Germany also reported the presence of EV genome in 56 out of 172 (32.6%) of these cases (Kühl et al., 2005a). In addition, 23 out of 245 (9.4%) patients with idiopathic left ventricular dysfunction were positive for EV genome (Kühl et al., 2005b). On the other hand, only 2% of patients (n = 125) with idiopathic DCM were positive for the EV genome in a study by Dennert et al in the Netherlands (Dennert et al., 2012).

A piece of evidence that may link CVB persistence to cardiac inflammation comes from a fatal case of myocarditis patient. The CVB2 genome, identified as a causative agent of myocarditis, had naturally occurring deletions within the cloverleaf motif (1–22 and 1–25 nucleotide deletions; Oka et al., 2005). These specific mutant viruses did not induce cytopathic effect in a mouse and human cardiac tissue, which is consistent with CVB2 persistence (Kim et al., 2005; Chapman et al., 2008; Smithee et al., 2015). Furthermore, Bouin et al reported persistent populations of CVB3 containing 15–48 deletions in the 5’UTR in human endomyocardial tissues (Bouin et al., 2016). It has been postulated that the lower replicative capacity of the virus harboring these deletions may enable its persistence in the cardiac muscle, and/or contribute to wild type CVB3 genomic recombination (Holmblat et al., 2014; Bouin et al., 2016). … In humans, assessing the mechanistic details involved in virus-induced disease pathology has been difficult due to limited sample availability. However, animal studies support the involvement of viruses in the disease process.

Badrinath A, Bhatta S, Kloc A. Persistent viral infections and their role in heart disease. Front Microbiol. 2022 Nov 24;13:1030440. doi: 10.3389/fmicb.2022.1030440.

I combine the last sentence of the Reviewer’s previous paragraph with the Reviewer’s next paragraph, since both address the same issue:

Not all myocarditis cases are caused by viruses but viral myocarditis has been shown to have autoimmunity.

Despite the statement in lines 93-95 and repeated on lines 667-669: “CVB3 only induces AM in mice when co‐inoculated with cardiac myosin or with heart antigens after being passaged through heart tissue (65-67).”  the literature on coxsackievirus B3 in mouse models of myocarditis going back to the 1960s has shown the presence of autoimmunity after viral infection.

There are a slew of different issues combined in the Reviewer’s comments, which I will need to unravel one by one. First, in the context of my assertion that myocardial proteins are required to induce myocarditis, the Reviewer states that Fairweather and Rose demonstrate myocarditis following virus infections:

For example: Fairweather D, Rose NR. Coxsackievirus-induced myocarditis in mice: a model of autoimmune disease for studying immunotoxicity. Methods. (2007) 41:118–22. doi: 10.1016/j.ymeth.2006.07.009;

Here is what Fairweather and Rose themselves state in that paper:

PAGE 119: “In the model of CVB3-induced myocarditis presented here, intraperitoneal (ip) inoculation of BALB/c mice with heart-passaged CVB3 (Nancy strain), which contains virus and cardiac myosin, induces inflammatory heart disease that is remarkably similar to disease induced by inoculation with adjuvant and cardiac myosin (experimental autoimmune myocarditis or EAM) [14–17].”

So, the Fairweather and Rose paper support my position, not the Reviewer’s.

The Reviewer also cites:

Latva-Hirvela J, et al. Development of troponin autoantibodies in experimental coxsackievirus B3 myocarditis. Eur J Clin Invest. (2009) 39:457–62. doi: 10.1111/j.1365-2362.2009.02113.x;

Takata S, et al. Identification of autoantibodies with the corresponding antigen for repetitive coxsackievirus infection-induced cardiomyopathy. Circ J. (2004) 68:677–82. doi: 10.1253/circj.68.677.

Here, I agree with the Reviewer that myocarditis was, indeed, induced without inoculating the animals with cardiac proteins. I have modified my language to be more accurate. However, I do not agree that these experiments (and those performed by many other investigators using the same IP methods) do not require presentation of myocardial proteins to the immune system. Most animals sustaining myocardial damage using the methods cited in the Latva-Hirvela paper die before the chronic (autoimmune) form of the disease can be manifested (see below). Thus, most papers using this method do not report how many animals were infected but only how many survived the procedure to go on to develop chronic myocarditis. These survivors are characterized by acute myocardial damage that reveals hidden cardiac antigens such as myosin that are immunologically processed along with the virus. Moreover, as I am sure the Reviewer is aware, the amount of virus necessary to elicit myocarditis in the survivors is often extraordinarily large (105-106 PFU) or, if lower, must be delivered to very young animals and the CVB strain must be tightly matched to the rodent strain (Gauntt C, Higdon A, Bowers D, Maull E, Wood J, Crawley R. What lessons can be learned from animal model studies in viral heart disease? Scand J Infect Dis Suppl. 1993;88:49-65). These features of the CVB-only model hardly match the age distribution or etiology of human myocarditis. (Indeed, the Reviewer makes a big deal below about the TCR-CVB matches in this study being generalizable over a very large number of CVB strains in order for my results to be of any value, yet this generalizability is not present in the CVB-only model of myocarditis!)

The CVB+cardiac-proteins method developed by Fairweather and Rose (see above) is far more efficient at causing myocarditis, never acutely lethal, and generalizable across a wider range of combinations of rodent-CVB strains – all characteristics of human autoimmune myocarditis. I quote from another Fairweather and Rose paper below concerning these issues and their hypothesis that coxsackie (and other) viruses are not the causes of autoimmune myocarditis, but adjuvants to cardiac proteins:

Fairweather D, Frisancho-Kiss S, Rose NR. Viruses as adjuvants for autoimmunity: evidence from Coxsackievirus-induced myocarditis. Rev Med Virol. 2005 Jan-Feb;15(1):17-27. doi: 10.1002/rmv.445.

Viral models that induce myocarditis by direct damage to the heart have not always been clearly distinguished

from viral models with a concurrent autoimmune pathogenesis. … It is important to note that the model of CB3-induced myocarditis used in our laboratory [coxsackievirus plus myocardial proteins] induces damage and inflammation in the heart largely by immune-mediated mechanisms rather than direct viral damage, since interpretation of the mechanisms involved in disease pathogenesis will be affected. In our model, CB3 replicates at a relatively low level in the heart and there are no deaths due to the virus in contrast to other virus induced

models of myocarditis [3]. In many other models, viral replication in the heart results in necrosis of myocytes and fibrosis during the early phase of myocarditis with few animals surviving to the chronic phase [34–36]. Since adult [human beings] rarely die from acute Coxsackievirus infections [37], it is felt that the low-dose model more closely resembles the disease as it occurs in human populations. … Furthermore, certain amyocarditic’ strains of CB3 are capable of replicating in the heart without causing inflammation [38], indicating that viral replication alone does not determine the development of myocarditis….

The best evidence that an active or persistent viral infection is not required for the development of

myocarditis comes from the demonstration that inoculation of BALB/c mice with cardiac myosin emulsified in the adjuvant CFA induces experimental autoimmune myocarditis (EAM) [41]. Importantly, the pathogenesis of EAM closely resembles the biphasic myocarditis associated with CB3 infection (Figure 2).

In short, there is obviously significant disagreement within the myocarditis field about what is a useful model and and is not necessary to induce disease. You have taken one position; I another. This disagreement is not something that can be resolved in the kind of back-and-forth we are having here, but rather only by allowing divergent hypotheses to be tested against each other. All I am asking is to allow my hypothesis to see the light of day so that it can be tested against yours.

What I was hoping to help you understand is that the two myocarditis cases from 1994 that you used for TCRs in the AM group most likely did have viral myocarditis because they were able to use monoclonal anti-coxsackievirus antibodies to detect coxsackievirus B antigens in the heart tissue. 

And I already explained in my previous response that antibodies cannot be used in this in the absence of direct evidence of virus genome to determine infection because these antibodies have been demonstrated to cross-react with cardiac antigens. Since antibodies raised against coxsackievirus bind to damaged cardiac tissue (the Reviewer admits that autoimmunity is present in their own comments!), and myocarditis is characterized by damaged cardiac tissue, this test cannot distinguish between the presence of virus and the presence of cardiac proteins. That’s the whole point about molecular mimicry of cardiac proteins for coxsackievirus antigens. Besides, as I pointed out before, the authors of the study under discussion did use RT-PCR to test for the presence of coxsackievirus genome and found none. The Reviewer cannot use their conjecture to invalidate the study’s experimental conclusion nor guess that there might have been replication deficient virus particles present.

I don’t agree that the detection of the virus in the heart tissue is background staining due to cross reactivity. The monoclonal antibodies used to detect the CVB antigens gave a strong signal in the myocarditis tissues and not in the normal tissues because these monoclonals were selected as antibodies which would not bind uninfected cardiac tissues.  The figures demonstrate this in Luppi et al. 2003 Human Immunology 64: 194-210.

Whether you agree or not is beside the point. My explanation takes into consideration the antibody data, the RT-PCR data and the presence of autoimmunity; yours ignores the RT-PCR data and the autoimmunity. Basic logic dictates that mine is the stronger interpretation because it explains more data.

Besides, let’s look at your next sentence:

  I agree that many cases of myocarditis may develop autoimmunity…

So, if many cases of myocarditis develop autoimmunity, how can one use antibodies that are cross-reactive with cardiac tissue to unambiguously identify the presence of coxsackievirus itself? The fact that the antibodies bind to myocarditis tissue but not normal cardiac tissue can just as easily be evidence of cardiac damage making cardiac antigens available to the antibodies as evidence for the presence of (hypothetical) virus. After all, what characterizes infected from non-infected tissue: tissue damage!

They probably were not using an RT-PCR capable of detecting the very low level of defective persisting coxsackievirus.

Neither of us knows how sensitive the RT-PCR was for defective coxsackievirus. The fact is that the paper reported that no viral genome was detectable. Using a hypothetical, as you do, to argue that virus might still be present is hardly a strong argument. Besides, probabilistically, your hypothetical defective virus was not even present (see above) and, even if it was, may or may not have contributed to the disease process (see above, both with reference to whether defective viruses actually contribute to myocarditis pathology and with reference to the Fairweather and Rose argument that non-inflammatory virus infection does not result in myocarditis).

and I agree that you are using two patients who have enteroviral myocarditis.  Very likely these patients have TCRs related to that infection which you have demonstrated to have reactivity to streptococcal antigens but again this is a very small number of patients. I’m not satisfied that you have sufficient support for your hypothesis using 2 patients who had a coxsackievirus infection of the heart and may or may not have had prior streptococcal infection. 

I agree that having TCR from only two patients is sub-optimal but I have stated that this is a serious limitation of the analysis in several places in the paper. I’m not trying to pull the wool over anyone’s eyes here: I’m trying to direct attention to a number of issues that are generally ignored, such as how coxsackieviruses can be associated with multiple autoimmune diseases and why TCR mimic different bacterial antigens in each disease.  These observation may, or may not, hold up to further investigation, but if they do hold up, they open up a whole bunch of very interesting questions and new directions for research. Moreover, every aspect of my hypothesis is testable and the object of this paper is to stimulate the research to necessary to validate or invalidate it. Progress isn’t made by saying we can’t move forward till convincing data are available because you can’t generate convincing data until you see a reason to do so.

I will restate the paragraph of my review that the author found difficult to interpret:

I appreciate that the difficulties of finding TCR sequences from patients with myocarditis or with type 1 diabetes limits the significance of the correlations that the author is drawing.  This is further complicated by the fact that the author is not using the data available to him in GenBank to demonstrate that he is defining the cross-reaction to antigens likely to be conserved in enteroviral infections.  If these viral epitopes were in the capsid of the virus, different enterovirus types would generate different TCRs and there would likely be a limited number of types which would be associated with myocarditis (if this mechanism played a large part in generating pathology).

Yes, just like the CVB-only model for inducing myocarditis in rodents! See above.

I see that the author has noted the viral protein in the second version of this manuscript but has not defined whether these are epitopes conserved or not conserved among the coxsackievirus Bs or enterovirus Bs.

This is simply false as can easily be verified by inspection of the previously revised manuscript! The revised Figures introduced in the previous version provided not only the protein names but also provide how many coxsackievirus A and B strains share the same protein sequences. In most cases, the few strains share the same sequence but in most cases, the number of strains is hundreds. Please note that I also included data on the conservation of the sequences for the Strep antigens as well. No, I did not attempt to do the same for all types of human enteroviruses, mainly because the number of matches quickly went into the thousands for many of the proteins, which not only became ungainly to try to estimate but also because the BLAST program has a limit on the number of best matches it will show, which was regularly exceeded. I have now put this limitation in the Methods.

The author has stated in the reply to my previous review: “I do think the fact that enterovirus antibody titers are higher in RHD patients does suggest at least recent infection…).  I do not see this in the literature. It is known that some murine monoclonal antibodies cross react with streptococcal antigens, cardiac valve proteins and coxsackievirus B capsids (Cunningham M, et al. 1992 PNAS 89 (4) 1320-1324). These monoclonal antibodies also are pathogenic to cardiac cells. Are any of the TCRs in the myocarditis patients mimicking the epitopes suggested in this 1992 study?

An interesting suggestion. But no. I compared the various sequences listed in the paper against the entire list of TCR utilized in both the diabetes and myocarditis studies, and did so using overlapping 5-mers as well as the whole peptide sequences, but got no matches. This result is not, perhaps, very surprising given the report by Neu, et al. (1987) that natural coxsackievirus-induced myocarditis does not produce myosin-reactive autoantibodies:

Neu N, Craig SW, Rose NR, Alvarez F, Beisel KW. Coxsackievirus-induced myocarditis in mice: cardiac myosin autoantibodies do not cross-react with the virus. Clin Exp Immunol 1987; 69: 566–574.

I have to guess that the results reported in Cunningham, et al (1992) are a result of inoculating the animals with purified peptides along with adjuvant, which likely results in a different immune response than from a natural virus infection.

I don’t see any evidence in references 59-62 that enteroviral antibodies are increased in rheumatic fever patients.

Why would you see such evidence? The context in which references 59-62 are cited is in a paragraph summarizing some of the bacteria associated with myocarditis/RHD: “If one considers rheumatic heart disease (RHD) as a form of autoimmune myocarditis, then there is also a strong association with group A and group G streptococci (GAS and GGS) (reviewed in [59-61]). Streptococcal vaccines are also (rarely) associated with induc-ing AM [62].” This passage has nothing to do with whether enteroviral antibodies are present in RHD. That evidence is summarized several paragraphs below in references 74-81.  Not every study of RHD has looked for the presence of enteroviruses along with strep and if the investigators didn’t look, they can hardly have reported whether such viruses were present! Case in point:

In the case described in Makaryus et al. (J Infect Dis 2006 14(4): e1-4) post streptococcal vaccination followed by viral (not necessarily enteroviral) upper respiratory infections caused recurrent myocarditis but no increase in viral or streptococcal antibodies was noted.

Wow! Using lack of evidence as evidence of absence! A classic logical fallacy. Well, yes, you are correct that, “no increase in viral or streptococcal antibodies was noted.” However, since this is because the authors of the Makaryus study never, at any point in the patient’s disease, measure antibody titers, the lack of this information does not in any way address the observation that significant titers of coxsackievirus and streptococcal antibodies were found in the papers I cited (refs 74-81). Sorry, but not measuring something doesn’t constitute evidence that the thing that needs to be measured is absent.

All in all, the hypothesis of this study is interesting but very limited.  If the author wishes to use this as a basis for proposing further studies, it help if the author made the significance of the figures (number of patients from which the TCRs were obtained, lack of a matched control group) and made the background of these diseases in the literature clear.  It is detracting from the value of this study to make erroneous claims about what is known of myocarditis and type 1 diabetes.

The many limitations of this study have been made clear in a very long and detailed “Limitations” section. As for the “erroneous claims”, I suggest for the many reasons given in this and my previous response, that the Reviewer, not I, is the one making them.

Comments on the Quality of English Language

There a quite a few typographic errors in the reference numbering.

Hope you have a sense of humor, because this point is pretty funny given that you just misspelled “are”! However, I have run spellcheck and hopefully found and corrected the typos.

Submission Date

09 November 2023

Date of this review

15 Jan 2024 23:26:55